# Searching Lottery Tickets in Graph Neural Networks: A Dual Perspective

**Kun Wang**[3]*, **Yuxuan Liang**[4]*, **Pengkun Wang**[3], **Xu Wang**[3], **Pengfei Gu**[3],
**Junfeng Fang**[3], **Yang Wang**[1,2,3]*
[1] Key Laboratory of Precision and Intelligent Chemistry, University of Science and Technology of China (USTC), Hefei, China
[2] School of Software Engineering, USTC. [3] School of Data Science, USTC.
[4] National University of Singapore, Singapore
{wk520529, pengkun, wx309, fjf, gpf9061}@mail.ustc.edu.cn,
angyan@ustc.edu.cn*, yuxliang@outlook.com*

## Abstract

Graph Neural Networks (GNNs) have shown great promise in various graph learning tasks. However, the computational overheads of fitting GNNs to large-scale graphs grow rapidly, posing obstacles to GNNs from scaling up to real-world applications. To tackle this issue, Graph Lottery Ticket (GLT) hypothesis articulates that there always exists a sparse subnetwork/subgraph with admirable performance in GNNs with random initialization. Such a pair of core subgraph and sparse subnetwork (called graph lottery tickets) can be uncovered by iteratively applying a novel sparsification method. While GLT provides new insights for GNN compression, it requires a full pretraining process to obtain graph lottery tickets, which is not universal and friendly to real-world applications. Moreover, the graph sparsification in GLT utilizes sampling techniques, which may result in massive information loss and aggregation failure. In this paper, we explore the searching of graph lottery tickets from a complementary perspective – transforming a random ticket into a graph lottery ticket, which allows us to more comprehensively explore the relationships between the original network/graph and their sparse counterpart. Compared to GLT, our proposal helps achieve a triple-win situation of graph lottery tickets with high sparsity, admirable performance, and good explainability. More importantly, we rigorously prove that our model can eliminate noise and maintain reliable information in substructures using the graph information bottleneck theory. Extensive experimental results on various graph-related tasks validate the effectiveness of our framework.

## 1 Introduction

Graph Neural Networks (GNNs) Kipf & Welling (2016); Hamilton et al. (2017) have recently emerged as the dominant model for a diversity of graph learning tasks, such as node classification Velickovic et al. (2017), link prediction Zhang & Chen (2019), and graph classification Ying et al. (2018). The success of GNNs mainly derives from a recursive neighborhood aggregation scheme, i.e., message passing, in which each node updates its feature by aggregating and transforming the features of its neighbors. However, GNNs suffer notoriously high computational overheads when scaling up to large graphs or with dense connections, since conducting message passing over large or dense graphs proves costly for training and inference Xu et al. (2018); You et al. (2020).

To alleviate such inefficiency, existing approaches mostly fall into two research lines – that is, they either *simplify the graph structure* or *compress the GNN model*. Within the first class, many studies Chen et al. (2018); Eden et al. (2018); Calandriello et al. (2018) have investigated the use of sampling to reduce the computational footprint of GNNs. These sampling-based strategies are usually integrated with mini-batch training schedule for local feature aggregation and updating. Another representative is graph sparsification techniques Voudigari et al. (2016); Zheng et al. (2020); Li et al. (2020b) which improve training or inference efficiency of GNNs by learning to remove redundant edges from input graphs. In contrast to simplifying the graph structure, there are much fewer prior studies on pruning or compressing GNNs Tailor et al. (2020), as GNNs are generally less parameterized than DNNs in other fields, e.g., computer vision Wen et al. (2016); He et al. (2017).

---

*Yang Wang and Yuxuan Liang are the corresponding authors.

Further, Graph Lottery Ticket hypothesis (GLT) Chen et al. (2021) has surprisingly killed two birds with one stone, i.e., for the first time it simultaneously simplifies the input graph and prunes the GNNs without compromising model performance. The key insight is to generalize the theory of Lottery Ticket Hypothesis (LTH) Frankle & Carbin (2018) to GNNs. Recall that LTH articulates there always exist sparse and high-performance subnetworks in a dense network with random initialization (like winning tickets in a lottery pool), GLT delineates a **Graph Lottery Ticket** as a *combination* of core subgraph and sparse subnetwork with admirable performance. More specifically, GLT first devises a unified GNN sparsification (UGS) strategy for *jointly* pruning the graph adjacency matrix as well as the network weights, and then iteratively applies UGS to uncover the winning tickets in GNNs. Extensive experiments on GNN benchmarks have verified the effectiveness of GLT across various architectures, learning tasks, and initialization ways.

After revisiting the theory of GLT, we expose two crucial factors that may impede GLT in practice. Firstly, GLT takes a whole pretraining process to obtain a sparse subnetwork, which limits its applicability to real-world usages and meanwhile complicates the investigation of the relationships between the original network (or graph) and their sparse counterparts. Secondly, the sampling-based graph simplification in GLT may lead to two devastating challenges: a) *Information loss:* pruning subgraph edges as GLT does may cause massive information loss, resulting in performance collapse Wu et al. (2022a). b) *Aggregation failure:* as the sparsity increases, some "unimportant" edges may be discarded by means of a pruning algorithm, but sometimes they connect two very important local communities (see Fig. 1).

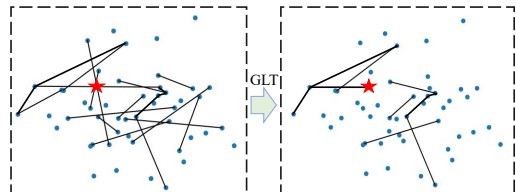

Figure 1: *Left*: graph before sparsification. *Right*: a sparse graph obtained by GLT Chen et al. (2021). The red star (★) denotes a node that connects two important communities. After adopting GLT, it can be seen that the edge connecting the two communities is discarded. Consequently, ★ can no longer aggregate information from both sub-structures.

In this paper, we investigate a more universal yet challenging problem from a complementary perspective of GLT: *how to transform a randomly selected ticket (i.e., a pair of graph and network) to a graph lottery ticket in GNNs?* Compared to the magnitude-based network pruning and graph sparsification in GLT, such a transformation process enables us to more comprehensively explore the relationships between the original network/graph and their sparse counterparts. Two-fold efforts are made by us to answer the above question, including **regularization-based network pruning** and **hierarchical graph sparsification**.

Primarily, we present the first attempt to generalize the Dual Lottery Ticket Hypothesis (DLTH) Bai et al. (2022) for GNN network pruning. Being initially designed for pruning deep neural networks, DLTH utilizes a Gradually Increased Regularization (GIR) term Wang et al. (2020a) to transfer the model expressivity from the discarded part to the remaining part. When adapting to GNNs, we first randomly select a target sparse subnetwork within the original dense network, and then attach GIR on the rest part to stimulate magnitude discrepancy among the parameters. In other words, as the regularization penalty factor increases, the information is continuously extruded from the rest part into our target subnetwork. Once the difference among parameters is discrepant enough, we remove the rest part to realize a final sparse network.

However, GIR is not applicable to graph sparsification when generalizing DLTH for finding graph lottery tickets. To this end, we propose *Hierarchical Graph Sparsification* (HGS) that is not only compatible with the GIR-based pruning strategy but also mitigates the information loss and aggregation failure issue in GLT. HGS learns a differentiable soft *assignment matrix* for nodes at each GNN layer, projecting nodes to a set of clusters which is then utilized as the coarsened input for the next GNN layer. Hierarchical representations of graphs are produced accordingly. Finally, we element-wise product the adjacency matrix of the coarsened graph at each GNN layer with a trainable mask for graph sparsification. In this way, useful information is extruded into the anticipative structure, thereby avoiding massive information loss. Note there are no node or edge dropping operations in our method, HGS can naturally remedy the aggregation failure in GNNs as well.

We elaborately unify the above regularization-based pruning and hierarchical graph sparsification into a single framework for transforming a random-selected ticket into a graph lottery ticket in GNNs, leading to a dual perspective of GLT. We therefore name our framework as **Dual Graph**

**Lottery Tickets** (DGLT). Similar to GLT, DGLT is model-agnostic and makes no assumptions on the graph structure, and can be easily applied and scaled up to a variety of graph-based learning tasks. To enhance its explainability, we theoretically prove our information extrusion approach from the popular Graph Information Bottleneck (GIB) theory. Our contributions are summarized as follows:

- We explore a new and non-trivial problem of transferring a random ticket to a graph lottery ticket in GNNs. Compared to GLT which pretrains dense GNNs to recognize graph lottery tickets, transferring a random ticket into a pair of high-performance sparse network and core subgraph is more appealing and valuable in practical usage, which allows us to investigate the relationships between the original network/graph and their sparse counterparts in a principle way.

- We present the Dual Graph Lottery Ticket (DGLT) framework to transform a random ticket into a *triple-win* graph lottery ticket, i.e., with high sparsity, high performance, and good explainability. DGLT prunes the GNN architecture by GIR-based information extrusion and sparsifies the input graph in a hierarchical manner to defeat information loss and aggregation failure in GLT. Moreover, the graph information bottleneck theory is utilized to guarantee the algorithm's preeminence.

- Extensive experiments are conducted on GNN benchmarks to examine our DGLT. The results show that DGLT consistently outperforms GLT across various graph/network sparsity over these benchmarks. For node classification, DGLT achieves $40\% \sim 85\%$ graph sparsity and $65\% \sim 92\%$ weight sparsity (no performance degradation), with about $13\% \sim 30\%$ sparsity improvement on graph and $5\% \sim 10\%$ weights sparsity improvement. For a large-scale dataset, i.e., Ogb-Collab, our model can obtain graph lottery tickets with nearly $43\%$ sparsity gain on the graph. These findings demonstrate its potential in a wide range of real-world applications.

## 2 PRELIMINARY & RELATED WORK

**Graph Neural Networks**. Given an undirected graph $\mathcal{G} = \{\mathcal{V}, \mathcal{E}\}$ with a node set $\mathcal{V}$ and an edge set $\mathcal{E}$, GNNs aim to learn a representation vector of a node or an entire graph based on the adjacency matrix $A \in \mathbb{R}^{|\mathcal{V}| \times |\mathcal{V}|}$ and node features $X \in \mathbb{R}^{|\mathcal{V}| \times F}$. Modern GNNs mostly follow a message passing strategy, in which we iteratively update the representation of a node $v_i \in \mathcal{V}$ by aggregating and transforming the representations of its neighbors. For example, Kipf & Welling (2016) propose a two-layer GNN with learnable parameters $\Theta = \{\Theta^{(0)}, \Theta^{(1)}\}$ for node classification as:

$$\mathcal{Z} = \mathcal{S}\left(\hat{A}\sigma\left(\hat{A}X\Theta^{(0)}\right)\Theta^{(1)}\right), \quad \text{and } \mathcal{L}\left(\mathcal{G}, \Theta\right) = -\sum\nolimits_{v_i \in \mathcal{V}_l} y_i \log\left(z_i\right) \text{ is the loss function,}$$

where $\mathcal{Z}$ is the prediction results, $\sigma\left(\cdot\right)$ denotes an activation function, $\hat{A} = \tilde{D}^{-\frac{1}{2}}\left(A + I\right)\tilde{D}^{-\frac{1}{2}}$ is the normalized adjacency matrix with self-loops and $\tilde{D}$ is the degree matrix of $A + I$. To optimize such GNNs, we minimize the cross-entropy loss $\mathcal{L}\left(\mathcal{G}, \Theta\right)$ over all labelled nodes $\mathcal{V}_l \subset \mathcal{V}$, where $y_i$ and $z_i$ represents the label and prediction of node $v_i$, respectively. More message passing schemes have been investigated in Hamilton et al. (2017); Velickovic et al. (2017); Li et al. (2020a).

Despite the promising results obtained by GNNs, they encounter notorious inefficiency when scaling up to large or dense graphs. Many streams of work have been dedicated to solving this issue. Graph sampling or sparsification accelerates the representation learning process of GNNs by manually or automatically extracting a sub-structure from the original graph Cheng et al. (2017); Chen et al. (2018); Rong et al. (2019); Li et al. (2020b); Faber et al. (2021). Graph compression algorithms attempt to merge an original graph to form a new small graph for fast representation learning Chakeri et al. (2016); Chiang et al. (2019). Tailor et al. (2020); Zhou et al. (2021); Chen et al. (2022) prune GNNs for speeding up reasoning. Recently, GLT Chen et al. (2021) has presented the first attempt to jointly sparsify the input graph and the GNN model, which significantly trims down the computational cost without compromising predictive accuracy. More importantly, GLT has opened up a novel research line to the graph learning community on which our framework is built.

**Lottery Ticket Hypothesis.** LTH articulates that a sparse and admirable subnetwork can be identified from a dense network by iterative pruning Frankle & Carbin (2018). LTH is initially observed in dense networks and is broadly found in many fields Evci et al. (2020); Frankle et al. (2020); Malach et al. (2020); Ding et al. (2021); Chen et al. (2020a; 2021); Sui et al. (2021). Derivative theories Chen et al. (2020b); You et al. (2021); Ma et al. (2021) are proposed to optimize the procedure of network sparsification and pruning. In addition to them, Dual Lottery Ticket Hypothesis (DLTH) considers a more general case to uncover the relationship between a dense network and its sparse counterparts Bai et al. (2022). It argues that when attaching GIR to a pre-selected part of a dense

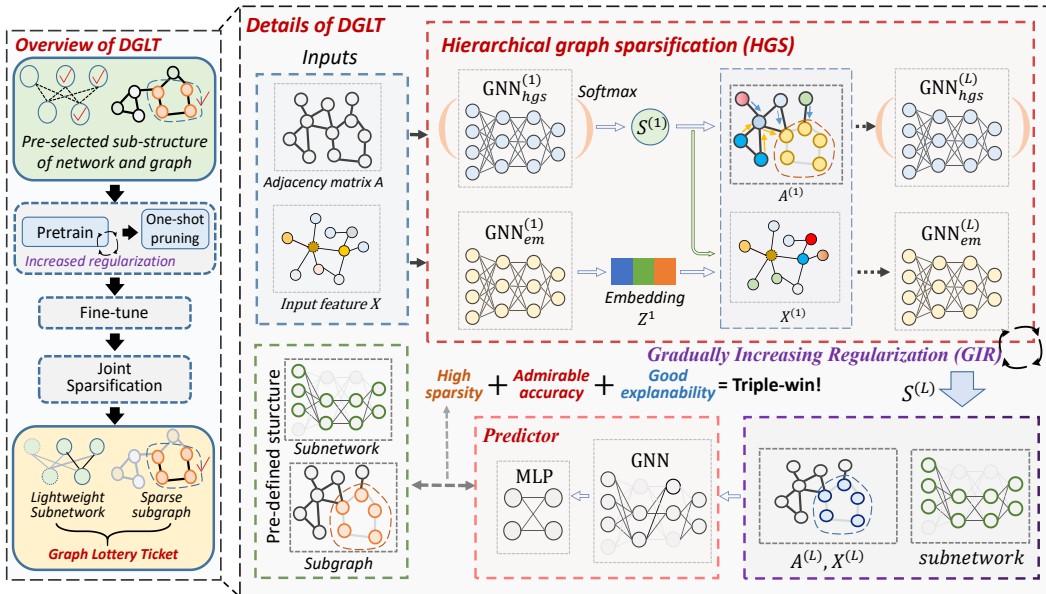

Figure 2: *(Left)* The overview of DGLT. *(Right)* The details of DGLT algorithm.

network, the complementary part can be transformed into an excellent winning ticket in an isolated training way. In this paper, we draw inspiration from DLTH and for the first time explore a dual problem of GLT, i.e., how to transform a random ticket into a graph lottery ticket in GNNs.

## 3 METHODOLOGY

Figure 2 presents an overview of our DGLT for transforming a random ticket to a graph lottery ticket in GNNs. The first step is selecting a target structure for subnetwork and subgraph. After that, the network is pretrained with Gradually Increased Regularization (GIR) for information extrusion. We meanwhile perform Hierarchical Graph Sparsification (HGS) to produce coarsened subgraph representations at each GNN layer. When GIR reaches a threshold, we stop adding penalty factors and train GNNs until the loss converges. Finally, we prune the network/graph for a joint sparsification with one shot and fine-tune the pruned model to obtain graph lottery tickets for evaluation. In the following parts, we commence by introducing HGS in Sec. 3.1 and then elaborate on how GIR benefits the sparse network training by extruding information from other weights to the target sparse structure (see Sec. 3.2). We further theoretically justify DGLT's power in transforming graph lottery tickets using the popular Graph Information Bottleneck (GIB) theory in Sec. 3.3. Frequently-used notations are listed in Appendix B for clarity.

### 3.1 HIERARCHICAL GRAPH SPARSIFICATION

Inspired by recent advances in clustering-based graph pooling methods Wu et al. (2022a); Ying et al. (2018); Roy et al. (2021), we propose Hierarchical Graph Sparsification (HGS) to generate hierarchical graph representations across different GNN layers for graph sparsification, where the number of nodes is reduced as the GNNs go deeper. Towards this goal, HGS learns a differentiable soft **assignment matrix** for nodes at each GNN layer, mapping input nodes to multiple clusters which are then fed to the next GNN layer as coarsened inputs.

As seen in Fig. 2, HGS learns the embedding representation $Z^{(l)}$ and assignment matrix $S^{(l)}$ at layer $l$ $(l = 1, 2, \ldots, L)$ via two non-shared GNNs layers, respectively. To be specific, we first adopt GNN sparsification layer (denoted as $\text{GNN}_{hgs}^{(l)}$) after each GNN embedding layer (denoted as $\text{GNN}_{em}^{(l)}$) for projecting nodes to a set of feature clusters $X^{(l)}$ and coarsened adjacency matrix $A^{(l)}$, which are then utilized as the coarsened input for the next GNN layer or final prediction. To facilitate subgraph generation, we impose a differentiable mask $m_A^{(l)}$ for $A^{(l)}$ in the hierarchical sparsification process:

$$Z^{(l)} = \text{GNN}_{em}^{(l)} \left( A^{(l-1)}, X^{(l-1)}; \Theta_{em}^{(l)} \right), \tag{1}$$

$$S^{(l)} = \text{softmax}\left(\text{GNN}_{hgs}^{(l)}\left(A^{(l-1)} \odot m_A^{(l-1)}, X^{(l-1)}; \Theta_{hgs}^{(l)}\right)\right), \tag{2}$$

For Eq. 1, embedding GNN at $l$-th layer applies $X^{(l-1)}$ and $A^{(l-1)}$ to produce node embedding representation $Z^{(l)}$ for clustering in next sparsification layer. For Eq. 2, $\odot$ is the element-wise product operation, softmax function is applied in a row-wise fashion. assignment matrix $S^{(l)} \in \mathbb{R}^{n_{l-1} \times n_l}$ $(n_{l-1} > n_l)$ can project coarsened subgraph into $n_l$ clusters. Notably that, in our implementation, we hope that $n_{l-1}$ is slightly larger than $n_l$ to ensure that graph information in the hierarchical clustering process extrudes steadily into small sub-structure. Given the embedding $Z^{(l)}$ and assignment matrix $S^{(l)}$, we apply the following equation to obtain adjacency matrix $A^{(l)}$ and embedding $X^{(l)}$ in the next layer:

$$A^{(l)} = S^{(l)^T} A^{(l-1)} S^{(l)}, \quad X^{(l)} = S^{(l)^T} Z^{(l)}. \tag{3}$$

After $L$ times of iterative embedding and sparsification of the input graph, we can obtain a resilient subgraph representation $\mathcal{G}_{hgs} = \{A^{(L)}, X^{(L)}\}$ in the last layer $L$, where $A^{(L)}$ denotes the connectivity between each pair of subgraph clusters and $X^{(L)}$ represents new node representation.

### 3.2 GRADUALLY INCREASED REGULARIZATION FOR INFORMATION EXTRUSION

$L_2$ regularization, commonly known as **weight decay** Loshchilov & Hutter (2018), is one of the most popular regularization terms. $L_2$ regularization draws the weights closer to the origin by adding a constraint term $\Omega(w) = \frac{1}{2}\alpha \|w\|_2^2$ to the loss function, where $\alpha$ represents the penalty coefficient. DLTH Bai et al. (2022) presents the first attempt to leverage $L_2$ regularization to extrude information from pre-selected part to its complementary counterpart. It demonstrates that when progressively increasing the penalty coefficient by adding a mini-step value, the difference between the weights will be separated and the unimportant weights naturally pushed to a position close to zero LeCun et al. (1989); Wang et al. (2020a) (see proof in Appendix A).

As depicted earlier, we get a sparse[1] representation $\mathcal{G}_{sub}$. Then, we directly element-product $A^{(L)}$ and a trainable mask $m_A^{(L)}$, and send a combination of $m_A^{(L)} \odot A^{(L)}$ with $X^{(L)}$ to GNN and MLP-layer predictors for label forecasting (Bottom right in Fig. 2). For DGLT framework, we pick a sparse network structure from the whole network parameters $\Theta$ and trainable matrices, then attach GIR on the rest part to extrude information toward the target structure. DGLT can be achieved as optimization following objective function:

$$\mathcal{L}_{DGLT} := \mathcal{L}\left(\{m_A \odot A_{all}, X\}, \Theta\right) + \xi \|m_A^*\|_2^2 + \rho \|\Theta^*\|_2^2 \tag{4}$$

$$\xi^{(p+1)} = \begin{cases} \xi^{(p)} + \xi_a & \xi^{(p)} < \xi_{ceil} \\ \xi^{(p)} & \xi^{(p)} = \xi_{ceil} \end{cases} \qquad \rho^{(p+1)} = \begin{cases} \rho^{(p)} + \rho_a & \rho^{(p)} < \rho_{ceil} \\ \rho^{(p)} & \rho^{(p)} = \rho_{ceil} \end{cases} \tag{5}$$

In Eq. 4, $m_A$ and $A_{all}$ denote the mask sets and adjacency matrices of all sparsification layers; in addition to the cross-entropy loss, the objective function contains two $L_2$ regularization terms, where $m_A^*$ and $\Theta^*$ are pre-selected parameters in $m_A$ and $\Theta$ which will be discarded after pretraining. Finally, under the interplay of progressively mini-step addition of regularization penalty, the $m_A^*$ and $\Theta^*$ are pushed to a position close to zero, and the information is extruded to the target part. In Eq. 5, $\xi^{(p)}$ and $\rho^{(p)}$ are the regularization terms at $p$-th updating. $\xi_a$ and $\rho_a$ indicate increased mini-step values of penalty coefficient. $\xi_{ceil}$ and $\rho_{ceil}$ indicate ceiling values of two regularization terms. We set range of regularization terms and control regularization value increases in a linear fashion until reaches their ceiling. To facilitate reading, we show the algorithm in Algo. 1

### 3.3 GIB VIEW OF GRADUALLY INCREASING REGULARIZATION

**Graph Information Bottleneck (GIB):** Information Bottleneck (IB), which originates from the information theory, aims to find a compression code of the input signals while retaining as much valid information as possible from the original encoding Tishby et al. (2000). In recent years, IB is naturally adapted to deep neural networks in a variety of applications and shows excellent effects Peng et al. (2018); Luo et al. (2019); Wang et al. (2020b); Wu et al. (2020); Yu et al. (2020); Miao et al. (2022). Our work builds upon graph field and often known as Graph Information Bottleneck (GIB),

---

[1]Different from pruning, we transform graphs into small size representations and note $1 - \frac{\|A^{(L)}\|_0}{|A|}$ as sparsity ratio. $\|\cdot\|_0$ and $|\cdot|_0$ are the number of non-zero elements and total number of elements, respectively.

as defined above, $Y = \{y_1, y_2 \ldots . y_{|\mathcal{V}|}\}$ denotes the label of all nodes and $\mathcal{G}$ is the input graph. GIB-based methods try to find an optimal subgraph $\mathcal{G}_s^*$ in subgraph set $\mathbb{G}_{sub}(\mathcal{G})$ by optimizing:

$$max_{\mathcal{G}_{sub} \in \mathbb{G}_{sub}(\mathcal{G})} I(\mathcal{G}_{sub}, Y) - \beta I(\mathcal{G}_{sub}, \mathcal{G}) \longrightarrow \mathcal{G}_s^* \qquad (6)$$

$\mathcal{G}_{sub}$ denotes a subgraph of the $\mathcal{G}$ and $I(\cdot)$ represents Shannon mutual information. $\beta$ is the hyperparameter used to control the proportion of the two parts. In Eq. 6, The first term is used to maximize the mutual information of subgraphs and labels, and the second term wants the subgraph to be as small as possible. GIB theory tries to identify unserviceable or noisy nodes of the training graph-structured data describe spurious correlation-versus-causations. Intuitively, a spurious correlation means that *after the introduction of nodes or graphs in the training set cannot increase or even reduce the mutual information between the training set and the label* Glymour et al. (2016); Arjovsky et al. (2019); Krueger et al. (2021). Different from sampling models, we try to transform a full graph into $\hat{\mathcal{G}}_s^*$ through a transformation function $T(*)$ (i.e., gradually increasing regularization) and guarantee spurious correlation removal (satisfies Eq. 6). We will provide theoretical analysis of how DGLT can obtain an admirable subgraph from GIB perspective.

**Lemma 1:** *when the penalty is increased at the same pace, because of different local curvature structures, the weights respond differently – weights with larger curvature will be less moved. As such, the magnitude discrepancy among weights will be magnified as regularization grows. Ultimately, the weights will naturally separate (unimportant weights tend to be very small and can be regarded as noise)* Wang et al. (2020a).

Based on **Lemma 1**, we list our two observations about spurious correlations distribution in graph: (1) Some nodes are pure noise; (2) Some node are composed of bootless correlations and useful associations. Under these two **Obs**ervations. We can get such solution: *Suppose each G contains subset $\mathbb{G}_{sub}(\mathcal{G})$, there exist $\mathcal{G}_s^* \in \mathbb{G}_{sub}(\mathcal{G})$ can remove spurious correlations, i.e., $I(\mathcal{G}_s^*, Y) \geq I(\mathcal{G}, Y)$. Unserviceable or noisy information are distributed in graph node features. In our implementation, we can transform a $\mathcal{G}$ to $\hat{\mathcal{G}}_s^*$ and make sure that $I(\hat{\mathcal{G}}_s^*, Y) \geq I(\mathcal{G}_s^*, Y) \geq I(\mathcal{G}, Y)$.*

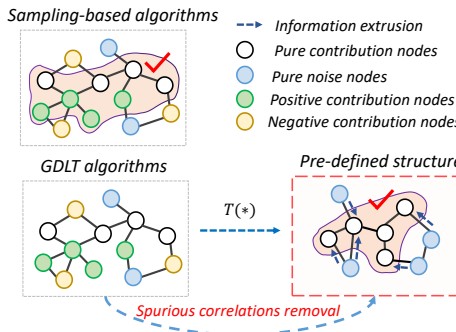

Figure 3: The white/blue node indicate that all features can increase/decrease $I(\mathcal{G}, Y)$, while the green/yellow nodes indicate that there are useful associations and false associations in the nodes, but in the end, the useful/spurious association dominates

**Theoretical Analysis. Obs 1: There exist pure noise nodes (note nodes set as $\dot{\mathcal{G}}_{sub}$) which make no contributions to $I(\mathcal{G}, Y)$ at all. Considering that our target is to maximize $I(\mathcal{G}_{sub}, Y)$:**

$$I(\mathcal{G}_{sub}, Y) = I(Y; \mathcal{G}, \mathcal{G}_{sub}) - I(Y; \mathcal{G}|\mathcal{G}_{sub}) = I(Y; \mathcal{G}) - I(Y; \mathcal{G}|\mathcal{G}_{sub}) \qquad (7)$$

Where the first equality is because *Chain Rule for Mutual Information* and the second equality is because $\mathcal{G}_{sub} \in \mathbb{G}_{sub}(\mathcal{G})$. $I(Y; \mathcal{G})$ is given from the beginning, so $I(Y; \mathcal{G})$ is a constant. Our target is to maximize $I(\mathcal{G}_{sub}, Y)$ (i.e., minimize the $I(Y; \mathcal{G}|\mathcal{G}_{sub})$). Since all nodes in $\dot{\mathcal{G}}_{sub}$ are noise, $I(\dot{\mathcal{G}}_{sub}, Y)$ in $I(\mathcal{G}, Y) - I(\mathcal{G}\backslash\dot{\mathcal{G}}_{sub}, Y) = I(\dot{\mathcal{G}}_{sub}, Y)$ reach the minimum. In GIB, $\mathcal{G}_s^* = \mathcal{G}\backslash\dot{\mathcal{G}}_{sub}$. For gradually increasing regularization, we can extrude information to a pre-defined structure. Since $\dot{\mathcal{G}}_{sub}$ are unserviceable, the amount of information squeezed is 0 in transformed $\hat{\mathcal{G}}_s^*$. Based on the above inference, we can obtain $I(\mathcal{G}_s^*, Y) = I(\hat{\mathcal{G}}_s^*, Y)$ in **Obs 1**.

**Theoretical Analysis. Obs 2: The spurious information is distributed in some nodes and these nodes are composed of valid information and spurious information.**

Suppose nodes set with spurious information is $\dot{\mathcal{G}}_{sub} = \{\dot{\mathcal{G}}_{sub}^*, \overline{\dot{\mathcal{G}}_{sub}^*}\}$, where nodes in $\dot{\mathcal{G}}_{sub}^*$ make positive contributions to $I(\mathcal{G}, Y)$ and $\overline{\dot{\mathcal{G}}_{sub}^*}$ makes no (even negative) contributions to $I(\mathcal{G}, Y)$. $I(\mathcal{G}, Y)$ reaches the maximum when $\mathcal{G}_s^* = \mathcal{G}\backslash\overline{\dot{\mathcal{G}}_{sub}^*}$. In our implementation, we extrude information to $\hat{\mathcal{G}}_s^*$ by training masks, and unimportant weights in masks tend to be exceedingly small and can be

regarded as noises. In this circumstance, $\hat{\mathcal{G}}_s^*$ contains at least more useful information than $\hat{\mathcal{G}}_s^*$ under the same sparseness as $\hat{\mathcal{G}}_s^*$. We can obtain $I\left(\hat{\mathcal{G}}_s^*, Y\right) > I\left(\mathcal{G}_s^*, Y\right) > I\left(\mathcal{G}, Y\right)$ in **Obs 2**.

To summarize, DGLT can obtain a comparable (even better) subgraph compare to the sampling-based algorithm. The difference between DGLT and other sampling methods is shown in Tab. 1

Table 1: Comparison between our DGLT, LTH Frankle & Carbin (2018), DLTH Bai et al. (2022) and GLT Chen et al. (2021). Pruning, graph controllability, network controllability, transformation, and pretrain denote the type of pruning process, if sparse graph structure is controllable, if sparse network structure is controllable, if the selected subnetwork needs transformation before fine-tuning, and if pretraining dense network is needed, respectively.

| Methods | Pruning | Graph controllability | Network controllability | Transformation | Pretrain |
|---------|---------|----------------------|------------------------|----------------|----------|
| LTH | Iterative | Infeasible in GNNs | $\times$ | No | Yes |
| DLTH | One-shot | Infeasible in GNNs | $\checkmark$ | Yes | No |
| GLT | Iterative | $\times$ | $\times$ | No | Yes |
| DGLT (ours) | One-shot | $\checkmark$ | $\checkmark$ | Yes | No |

## 4 EXPERIMENTS

In this section, we conduct extensive experiments to answer the following research questions: **RQ1:** How effective is our proposed DGLT algorithm in transforming a random ticket to a pair of sparse subnetwork and subgraph? **RQ2:** Can DGLT scale up to large-scale datasets? **RQ3:** Under a certain sparse ratio of subgraph (or subnetwork), how does the other ratio affect the model performance? **RQ4:** How does the proposed hierarchical graph sparsification impact the results? **RQ5:** How should we set the mini-step values of the increased regularization force?

### 4.1 EXPERIMENTAL SETTINGS

**Datasets.** Six benchmarks for GNN evaluation are employed in this paper to verify the effectiveness of our DGLT. To be specific, we choose three popular graph-based datasets, including Core, Citeseer and PubMed Kipf & Welling (2016) for node classification and link prediction. To test the scalability of DGLT, we further use a large-scale dataset called Ogbl-Collab Hu et al. (2020) for link prediction. Finally, we examine our algorithm for graph classification on D&D Dobson & Doig (2003) and ENZYMES Borgwardt et al. (2005). The statistics of these datasets can be seen in Table 5.

**Backbones & Parameter Settings.** For all selected backbones and datasets, we compare our DGLT algorithm with GLT Chen et al. (2021) and the random pruning algorithm under the same network settings. For regular-scale datasets, we adopt GCN Kipf & Welling (2016), GIN Xu et al. (2018) and GAT Veličković et al. (2017) as backbones. For Cgbl-Collab which is a large-scale dataset, we take 28-layer deep ResGCNs Li et al. (2020a) as our backbone for link prediction. To evaluate our DGLT on the graph classification task, GraphSAGE Hamilton et al. (2017) is leveraged as the backbone on D&D and ENZYMES. More details about experimental settings can be found in Appendix E.

### 4.2 CAN DGLT FINDS GRAPH LOTTERY TICKETS? (RQ1)

To answer **RQ1**, we compare our DGLT with GLT and random pruning on node classification. When investigating how accuracy changes with the growth of graph sparsity, we fix the weight sparsity to zero for stability, and vice versa. The results on Citeseer and PubMed are depicted in Fig. 4, and those on Cora is shown in Fig. 8. From Fig. 4 and 8, we have the following observations:

(1) **DGLT consistently outperforms GLT and random pruning** under the same graph/weight sparsity over all datasets, verifying the superiority of transforming a random ticket to a graph lottery ticket via information extrusion and hierarchical graph sparsification. For example, the graph lottery ticket[2] on PubMed+GIN identified by DGLT is with 85% graph sparsity or 87% weight sparsity. For Citeseer+GIN, we can obtain it with 65.0% and 91.0% sparsity on graph/weights using DGLT. As for random pruning, the effectiveness of the model decreased significantly when the sparsity rate increased, which further demonstrate the superiority of our DGLT algorithm.

(2) **DGLT for the first time enables us to search an ultra-lightweight subnetwork in GNNs.** For node classification on Cora and Citeseer, the model surprisingly shows no significant performance

---

[2]In our experiments, we specify a graph lottery ticket as a pair of subgraph and subnetwork with comparable accuracy to the baseline of full GNNs on full graphs.

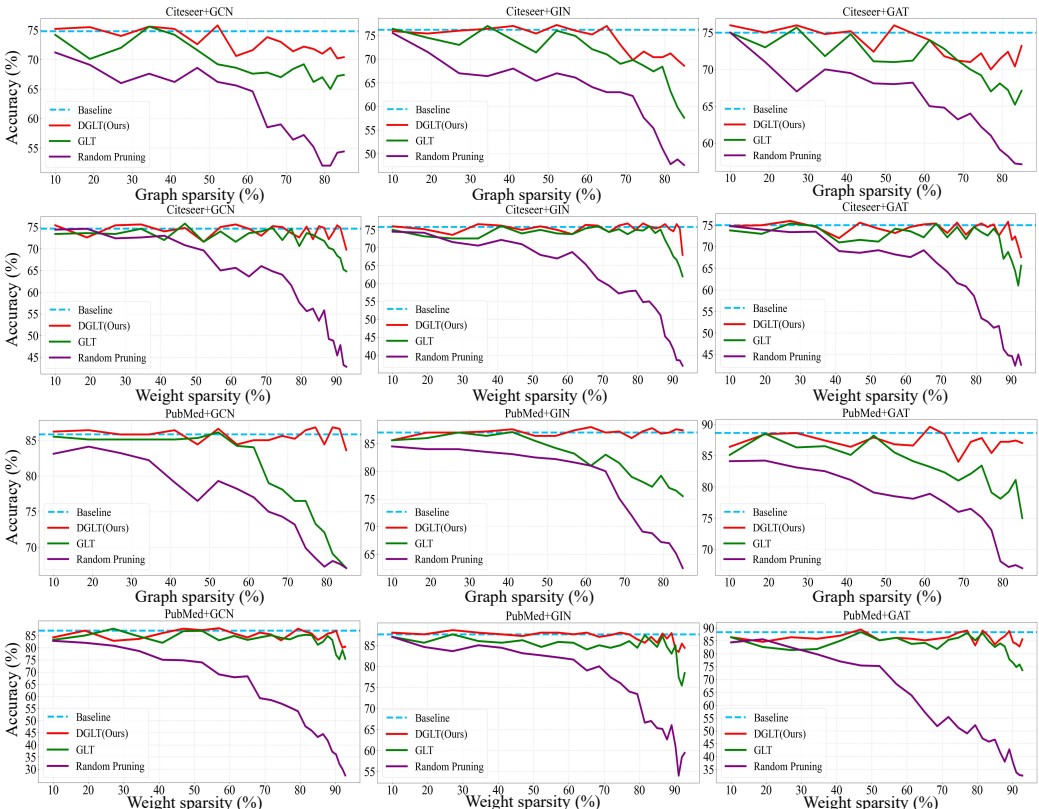

Figure 4: Results of node classification over Citeseer/PubMed with GCN/GIN/GAT backbones. Blue dash lines represent the baseline performance of full GNNs on full graphs.

drop (even surpassing the baseline performance) until 90% weight sparsity. For a larger dataset (i.e., PubMed), DGLT achieves 85.0% graph sparsity without performance degradation. We demonstrate that DGLT transforms graph lottery tickets by GIR, which can remedy the information loss and thus transfer a more informative pair of subgraph and subnetwork.

(3) **Whether we can find an extremely sparse subgraph depends on the property of input graphs.** Though DGLT can help find extreme sparse subgraphs in GNNs (e.g., on PubMed+GIN or PubMed+GAT), this phenomenon does not occur in smaller datasets (e.g., Cora). We argue that the graph property such as graph size may be an crucial factor to the possibility of transforming a very sparse subgraph. This question will be discussed further in Sec. 4.3. Besides node classification, we present additional empirical results on link prediction in Appendix F.

## 4.3    HOW DOES DGLT PERFORM ON LARGE-SCALE DATASET (RQ2)

To answer **RQ2**, we conduct experiments on the Ogbl-Collab dataset using ResGCNs as the backbone. We show the potential of DGLT in practice by evaluating the trade-off among the reasoning time overhead, accuracy, and memory savings. As shown in Fig. 5, DGLT performs better than GLT and random pruning across different graph sparsity and

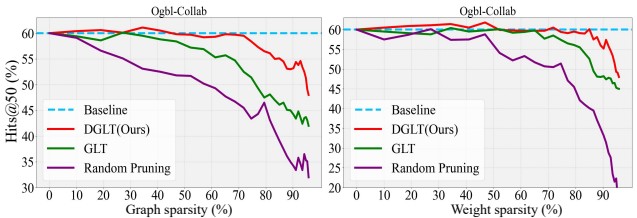

Figure 5: Link prediction results on Ogbl-Collab+ResGCNs.

weight sparsity. By using DGLT, we can obtain a graph lottery ticket with nearly 70% graph sparsity and 85% weight sparsity which outperforms GLT by 6%~7%. Meanwhile, we find that DGLT is more stable on large-scale datasets. Similar to DGLT's performance on PubMed, as graph sparsity increases (until ~90% sparsity), we witness a slow decline in performance, but there was a very obvious fluctuation in the small datasets (e.g. Cora), which shows that DGLT is more stable and conducive to expanding to large-scale datasets.

## 4.4 ABLATION STUDIES (RQ3 & RQ4)

To complement the experiments in Sec. 4.2, we investigate a more complex case described in **RQ3** by controlling the graph sparsity $P_g$ or weight sparsity $P_\theta$ at a fixed ratio (10%, 30%, 50%, and 70%) and examining the model performance under different sparsity of the other term. The experimental results are reported in Fig. 6. It can be seen easily that each line has a similar trend in both sub-figures, where the accuracy *slightly* drops as the growth of the graph (or weight) sparsity. For example, given $P_\theta = 70\%$, we can obtain an admirable subgraph (with only 2% lower accuracy) under 40% graph sparsity. The sparsified GNNs preserve excellent performance (only 8%~14% decrease on accuracy) even when the sparsity reaches 90%. This indicates the capability of DGLT in transforming graph lottery tickets with comparable performance to the original backbone. Moreover, we find that the lines in the right figure are more easily distinguished. When $P_g = 30\%$, an expressive subnetwork is achieved with 70% weight sparsity of the original network.

To evaluate hierarchical graph sparsification (HGS), we compare DGLT with its variant without HGS which attaches a trainable mask to the adjacency matrix at the last GNN layer for graph sparsification like GLT. As depicted in Tab. 2, DGLT surpass its variant without HGS by a large margin under the same graph sparsity. Such merits stem from its hierarchical information extrusion strategy, which allows the input graph to transfer information to the final small-size graph in a gradually compressed form and thus avoids the instability of one-shot rough extrusion.

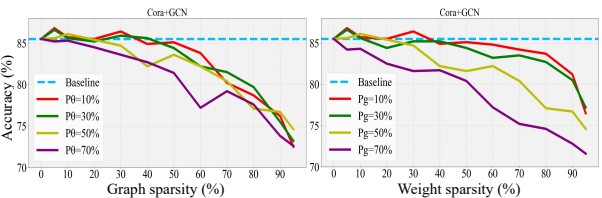

| Settings | | Graph Sparsity | | | | |
|---|---|---|---|---|---|---|
| Dataset | +HGS? | 10% | 20% | 40% | 60% | 80% |
| Cora | ✓ | 85.7 | 85.2 | 84.5 | 83.1 | 82.5 |
| +GCN | × | 81.1 | 81.4 | 80.9 | 78.1 | 71.2 |
| Citeseer | ✓ | 75.2 | 75.7 | 75.1 | 74.9 | 70.5 |
| +GIN | × | 74.1 | 71.8 | 69.5 | 67.4 | 65.3 |
| PubMed | ✓ | 87.5 | 90.4 | 86.6 | 88.1 | 87.4 |
| +GAT | × | 85.0 | 83.1 | 77.5 | 74.3 | 72.6 |

Figure 6: Effects of different pruning ratios for transforming sparse graphs and networks on Cora+GCN setting.

Table 2: Effects of HGS over different datasets and backbones.

## 4.5 MINI-STEP VALUES OF REGULARIZATION (RQ5)

We control graph/weight sparsity to 30%/70% and choose multiple mini-step values of regularization ($\xi_a$ and $\rho_a$ take the same mini-step values) for comparison. From Tab. 3, we argue that *the mini-step values of regularization should be maintained in a small regime.* When the progressive regularization force maintain in larger regime (>5e-4), the model effect decreases significantly. We demonstrate that the process of information extrusion should be slow and excessive steps of increased regularization may cause the model to fall into ill-condition.

Table 3: Different mini-step values of regularization over Cora/Citeseer/PubMed with different backbones. For clarity, the highest/second-highest performances are emphasized with red/blue fonts.

| Settings | | Mini-steps of Regularization | | | | | | |
|---|---|---|---|---|---|---|---|---|
| Datasets | Methods | 0 | 1e-6 | 5e-6 | 1e-5 | 5e-5 | 1e-4 | 5e-4 |
| Cora | GCN (baseline=86.72%) | 84.93±0.11 | 86.75±0.08 | 86.18±0.15 | 85.16±0.24 | 84.52±0.17 | 82.14±0.28 | 81.16±0.27 |
| | GIN (baseline=85.91%) | 84.17±0.08 | 85.41±0.14 | 85.01±0.22 | 84.84±0.17 | 84.07±0.15 | 82.68±0.15 | 80.13±0.14 |
| | GAT (baseline=85.43%) | 83.62±0.22 | 85.41±0.23 | 85.82±0.17 | 85.03±0.19 | 83.22±0.22 | 81.72±0.18 | 80.25±0.34 |
| Citeseer | GCN (baseline=75.73%) | 74.73±0.13 | 75.67±0.17 | 76.22±0.23 | 76.78±0.45 | 75.38±0.46 | 71.74±0.47 | 71.89±0.65 |
| | GIN (baseline=76.34%) | 74.54±0.33 | 76.27±0.22 | 76.07±0.38 | 73.44±0.25 | 74.47±0.28 | 72.17±0.34 | 71.09±0.65 |
| | GAT (baseline=76.65%) | 75.27±0.18 | 76.67±0.24 | 75.81±0.23 | 74.93±0.35 | 74.83±0.18 | 72.82±0.47 | 70.88±0.72 |
| PubMed | GCN (baseline=85.11%) | 84.25±0.13 | 85.17±0.24 | 85.74±0.25 | 83.24±0.22 | 83.16±0.46 | 81.87±0.17 | 79.82±0.14 |
| | GIN (baseline=84.84%) | 84.01±0.11 | 84.84±0.17 | 84.35±0.13 | 83.12±0.55 | 82.83±0.25 | 81.12±0.34 | 80.44±0.47 |
| | GAT (baseline=85.73%) | 84.08±0.14 | 85.67±0.17 | 85.07±0.21 | 83.58±0.34 | 82.77±0.37 | 81.05±0.28 | 79.89±0.45 |

## 5 CONCLUSION

In this paper, we propose a novel framework entitled Dual Graph Lottery Ticket (DGLT) that couples hierarchical graph sparsification and gradually increasing regularization to achieve triple-win graph lottery tickets (with high sparsity, admirable performance, and good explainability). Our work first points out that an admirable subgraph can be obtained by efficient hierarchical compression, which helps defeat the off-and-shelf sampling-based GNNs methods. We further generalize the key idea of the dual lottery tickets hypothesis for GNNs across various GNN backbones, learning tasks, and benchmarks. These explorations provide us a new perspective to uncover the relationships between the full model/graph and its sparse counterpart.

## 6 ACKNOWLEDGEMENT

This work is partially supported by the National Natural Science Foundation of China (No.62072427, No.12227901), the Project of Stable Support for Youth Team in Basic Research Field, CAS (No.YSBR-005), Academic Leaders Cultivation Program, USTC.

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

## A  PROOF OF GRADUALLY INCREASING REGULARIZATION

Regularization is long deemed as a tool on limiting the capacity of deep learning networks, by adding a penalty term $\Omega\left(\Theta\right)$ to the objective function $J$. We denote the regularized objective loss function by $\tilde{J}$:

$$\tilde{J}\left(\Theta; X, \mathrm{y}\right) = J\left(\Theta; X, \mathrm{y}\right) + \alpha\Omega\left(\Theta\right) \tag{8}$$

Where $\alpha \in [0, \infty)$ is a hyperparameter that weights the relative contribution of the penalty term. Setting $\alpha$ to 0 results in no regularization. Larger values of $\alpha$ correspond to more regularization. For $L_2$ regularization (commonly known as weight decay), $\Omega\left(\Theta\right) = \frac{1}{2}\left\|w\right\|_2^2$ is added into objective function. To simplify the presentation, we assume no bias parameter, so $\Theta$ is just equaled to $w$ and we can list objective function as:

$$\tilde{J}\left(\Theta; X, \mathrm{y}\right) = J\left(\Theta; X, \mathrm{y}\right) + \frac{\alpha}{2}w^T w \tag{9}$$

Where the gradient of objective function is:

$$\nabla_w \tilde{J}\left(\mathrm{w}; X, \mathrm{y}\right) = \alpha w + \nabla_w J\left(\mathrm{w}; X, \mathrm{y}\right) \tag{10}$$

The whole network can be optimized by:
$$w \leftarrow w - \epsilon\left(\alpha w + \nabla_w J\left(w; X, y\right)\right) \tag{11}$$

Written another way, the update is:
$$w \leftarrow (1 - \epsilon\alpha)w - \nabla_w J\left(w; X, y\right) \tag{12}$$

Further we simplify the analysis by making a quadratic approximation to the objective function in the neighborhood of the weights that obtains minimal unregularized training cost, $w^* = argmin_w\left(w\right)$. In the neighborhood of $w^*$, we can obtain:

$$J\left(\Theta; X, \mathrm{y}\right) = J\left(w^*\right) + \frac{1}{2}(w - w^*)^T H\left(w - w^*\right) \tag{13}$$

Where $H$ is the Hessian matrix of $J$ with respect to $w$ evaluated at $w^*$. There is no first-order term in this quadratic approximation, because $w^*$ is defined to be a minimum, where the gradient vanishes. Meanwhile, due to the $w^*$ is the minimum of $J$, we can conclude that $H$ is positive-semidefinite. The minimum of $\tilde{J}$ occurs where its gradient:

$$\nabla_w \widetilde{J}(w) = H(w - w^*) = 0 \tag{14}$$

**Gradually increasing regularization (GIR):** We add weight decay gradient in 14, as regularized $\tilde{J}$ reaches minimum (corresponding to $\tilde{w}$), we can obtain:

$$\alpha \tilde{w} + H(\tilde{w} - w^*) = 0 \implies \tilde{w} = (H + \alpha I)^{-1} H w^* \tag{15}$$

After increasing the penalty $\alpha$ by $\delta\alpha$, the new converged weights $\hat{w}$ have the same relation (15) with previous round convergence point $\bar{w}$:

$$\hat{w} = (H + \delta\alpha I)^{-1} H \bar{w} \tag{16}$$

where $I$ represents identity matrix.

For ease of exposition and in order to make the paper more self-contained, we prudently inherit and place the proofs which analysis in Wang et al. (2020a), we list two simplified cases to move forward. (1) $H$ is diagonal. For $\hat{w}_i$ with second derivative $h_{ii}$. As adding $\delta\alpha > 0$ regularized force, the new converged weights can be proved to be

$$\hat{w}_i = \frac{h_{ii}}{h_{ii} + \delta\alpha} \bar{w}_i \implies \frac{\hat{w}_i}{\bar{w}_i} = \frac{1}{\delta\alpha/h_{ii} + 1} \tag{17}$$

Where $\frac{\hat{w}_i}{\bar{w}_i} \in [0, 1)$ since $h_{ii} \geq 0$ and $\delta\alpha > 0$. As seen, when $h_{ii} \uparrow$, make $\frac{\hat{w}_i}{\bar{w}_i} \uparrow$, we can find that the weight relatively less moves towards the origin.

(2) We consider a general case (2-d) instead of diagonal matrix of $H$. $\bar{w} = \left\{ \begin{array}{c} \bar{w}_1 \\ \bar{w}_2 \end{array} \right\}$, $H = \left( \begin{array}{cc} h_{11} & h_{12} \\ h_{21} & h_{22} \end{array} \right)$, $\hat{H} = \left( \begin{array}{cc} h_{11} + \delta\alpha & h_{12} \\ h_{21} & h_{22} + \delta\alpha \end{array} \right)$. After adding mini-step value of regularization,

$$\left\{ \begin{array}{c} \hat{w}_1 \\ \hat{w}_2 \end{array} \right\} = \frac{1}{\left| \hat{H} \right|} \left\{ \begin{array}{c} \left( h_{11} h_{22} + h_{11} \delta\alpha - h_{12}^2 \right) \bar{w}_1 + \delta\alpha h_{12} \bar{w}_2 \\ \left( h_{11} h_{22} + h_{22} \delta\alpha - h_{12}^2 \right) \bar{w}_2 + \delta\alpha h_{12} \bar{w}_1 \end{array} \right\} \approx \frac{1}{\left| \hat{H} \right|} \left\{ \begin{array}{c} \left( h_{11} h_{22} + h_{11} \delta\alpha - h_{12}^2 \right) \bar{w}_1 \\ \left( h_{11} h_{22} + h_{22} \delta\alpha - h_{12}^2 \right) \bar{w}_2 \end{array} \right\} \tag{18}$$

$$\implies \frac{\hat{w}_1}{\bar{w}_1} = \frac{1}{\left| \hat{H} \right|} \left( h_{11} h_{22} + h_{11} \delta\alpha - h_{12}^2 \right) \quad \frac{\hat{w}_2}{\bar{w}_2} = \frac{1}{\left| \hat{H} \right|} \left( h_{11} h_{22} + h_{22} \delta\alpha - h_{12}^2 \right) \tag{19}$$

As seen, we can draw the same conclusion: $h_{11} > h_{22}$ also results in $\frac{\hat{w}_1}{\bar{w}_1} > \frac{\hat{w}_2}{\bar{w}_2}$. According to Wang et al. (2020a), when the penalty is increased at the same pace, due to the different local curvature structure, the weights response in a different way. Weights with larger curvature will be less moved. As regularization gradually increases, the magnitude discrepancy among weights will be magnified and the unimportant weights tend to zero and can be regarded as noise.

## B  NOTATIONS

For the convenience of reading and consulting, we put all the notations of this work in Table 4.

## C  ALGORITHM OF DUAL GRAPH LOTTERY TICKET FRAMEWORK

In this part, we summarize our DGLT algorithm process in Algo. 1. We first select a part of parameters to be discarded later. Then we initial our framework parameters. and perform a backpropagation algorithm to update the network parameters. After that we adopt GIR for information extrusion. until the useful information is sufficiently extruded into the target structure, we one-shot pruning pre-defined structure and fine-tune unpruned parameters for model evaluation.

Table 4: The notations that are commonly used in Methodology (Sec. 3).

| Notation | Definition |
| --- | --- |
| $\mathcal{G} = \{\mathcal{V}, \mathcal{E}\}$ | Input graph |
| $A$ | Input adjacency matrix |
| $X$ | Input features |
| $Z^{(l)}$ | Embedding representation after $l$-th GNN embedding layer |
| $S^{(l)}$ | Assignment matrix for $l$-th GNN embedding layer output |
| $A^{(l)}$ | Adjacency matrix after $l$-th GNN sparsification layer |
| $X^{(l)}$ | Nodes representation after $l$-th GNN sparsification layer |
| $\Theta_{em}^{(l)} \; (l = 1, 2 \ldots L)$ | Parameters of layer $l$-th embedding GNN |
| $\Theta_{hgs}^{(l)} \; (l = 1, 2 \ldots L)$ | Parameters of layer $l$-th sparsification GNN |
| $m_A^{(l)}$ | Trainable matrix for masking $A^{(l)}$ |
| $A_{all}$ | $A_{all} = \left\{ A, A^{(1)} \ldots A^{(L)} \right\}$ represent all adjacency matrix outputs |
| $m_A$ | $m_A = \left\{ m_A^{(0)}, m_A^{(1)} \ldots m_A^{(L)} \right\}$ represent mask set for $A_{all}$ |

## D    COMPLEXITY ANALYSIS OF GLT AND DGLT

Follow the GLT Chen et al. (2021), we we present the complexity of DGLT algorithm. As for GLT, the inference time complexity of GLTs is $\mathcal{O}\left( L \times ||m_g \odot A||_0 \times F + L \times ||m_\theta||_0 \times |\mathcal{V}| \times F^2 \right)$, where $L$ is the number of layers, $||m_g \odot A||_0$ is the number of remaining edges in sparse graph, $F$ is the dimension of feature and $|\mathcal{V}|$ is the number of nodes. The inference time complexity of DGLT is $\mathcal{O}\left( ||m_A \odot A_{all}||_0 \times F + ||m^*||_0 \times |\mathcal{V}| \times F^2 \right) + \mathcal{O}\left( \mathcal{K} \right)$, where $m_A = \left\{ m_A^0, m_A^1 \ldots m_A^L \right\}$ represent mask set for all adjacency matrix outputs $A_{all}$. $F$ is the dimension of feature and $|\mathcal{V}|$ is the number of nodes. $||m^*||_0$ represents all remained parameters of two non-shared GNNs. $\mathcal{O}\left( \mathcal{K} \right)$ represents inference time complexity of learning the node embeddings and the assignment matrix. They are obtained by multiplying multiple matrices and the inference time complexity of $\mathcal{O}\left( \mathcal{K} \right) = \mathcal{O}\left( L \times |\mathcal{V}|^3 + L \times |\mathcal{V}| \times F \right)$.

## E    EXPERIMENTAL SETTINGS

**Metrics**: Accuracy is the proportion of correct prediction results in all predictions. ROC-AUC (Receiver Operating Characteristic-Area Under the Curve) value is equivalent to the probability that a randomly chosen positive example is ranked higher than a randomly chosen negative example.

---

**Algorithm 1** Dual Graph Lottery Tickets (DGLT) Algorithm (aligned with Fig. 2)

---

**Require:** Input graph $\mathcal{G} = \{A, X\}$, GNN $f\left( \mathcal{G}; m_A, \Theta \right)$ with initialization parameters $\Theta_0$ and $\mathrm{m}_{A_0}$, step size $\eta$, penalty hyper-parameters $\xi$ and $\rho$ of regularization terms.
 1: Select a part of parameters to be discarded later, i.e., $m_A^*$ in $m_A$ and $\Theta^*$ in $\Theta$.
 2: **for** layer $\phi = 1, 2 \ldots L$ in embedding GNN and sparsification GNN **do**
 3:     Learn $\phi$-layer embedding representation $\mathrm{Z}^{(\phi)}$ in Eq. 1.
 4:     Learn $\phi$-layer coarsened representation by $\mathrm{S}^{(\phi)}$ in Eq. 2.
 5:     Hierarchical cluster to form new graph in Eq. 3.
 6: Obtain $\mathcal{G}_{sub} = \{A^{(L)}, X^{(L)}\}$.
 7: $m_A^{(L)} \odot A^{(L)}$ and send $\mathcal{G}'_{sub} = \left\{ m_A^{(L)} \odot A^{(L)}, X^{(L)} \right\}$ into predictor.
 8: **for** iteration $i = 1, 2 \ldots E$ **do**
 9:     Forward $f\left( \{m_A \odot A, X\}, \Theta \right)$ to compute loss $\mathcal{L}_{DGLT}$ in Eq. 4.
10:     Update $\Theta_{i+1} \longleftarrow \Theta_i - \eta \nabla_{\Theta_i} \mathcal{L}_{DGLT}$
11:     Update $m_{A_{i+1}} \longleftarrow m_{A_i} - \eta \nabla_{m_{A_i}} \mathcal{L}_{DGLT}$
12:     Increase regularization penalty $\Theta_0$ and $\mathrm{m}_{A_0}$ by mini-step in Eq. 5
13: One-shot pruning of $m_A^*$ and $\Theta^*$.
14: Fine-tune the model with rest parameters.
15: **return** Dual graph lottery ticket.

---

Table 5: Dataset details. The description of the metrics is placed in Appendix E

| Dataset | Task | #Graphs | #Nodes | #Edges | #Features | #Classes | Evaluation Metric |
|---|---|---|---|---|---|---|---|
| Cora | Node classification/Link prediction | – | 2,708 | 5,429 | 1,443 | 7 | Accuracy/ROC-AUC |
| Citeseer | Node classification/Link prediction | – | 3,327 | 4,732 | 3,703 | 6 | Accuracy/ROC-AUC |
| PubMed | Node classification/Link prediction | – | 19,717 | 88,338 | 500 | 3 | Accuracy/ROC-AUC |
| Ogbl-Collab | Link prediction | – | 235,868 | 1,285,465 | 128 | 2 | Hits@50 |
| D&D | Graph classification | 1,178 | 334,925 | 16,886,092 | 89 | 2 | Accuracy |
| ENZYMES | Graph classification | 600 | 19,580 | 745,654 | 3 | 6 | Accuracy |

Hit@50 means that taking the candidate edge of the top 50, the proportion of the 50 edges is predicted correctly.

**Sparsity ratio**: we transform graphs into small size representations and note $1 - \frac{\left\|A^{(L)}\right\|_0}{|A|}$ as graph sparsity ratio. $\|\cdot\|_0$ and $|\cdot|_0$ are the number of non-zero elements and total number of elements, respectively. Similarly, weight sparsity denotes that the ratio of the discarded parameters to the total parameters in the whole network.

**Train-val-test Splitting of Datasets.** To rigorously verify the effectiveness of our proposed DGLT algorithm, we control the network designs consistent under the same task. As for node classification task of regular-size datasets, we follow the same data split criteria among different backbones, i.e., 700 (Cora), 420 (Citeseer) and 460 (PubMed) labeled data for training, 500 nodes for validation and 500 nodes for testing. As for link prediction, we shuffle the datasets and sample 85% edges for training, 10% for validation, 5% for testing, respectively. For Ogbl-Collab, in order to simulate a real collaborative recommendation application, we take the cooperation before 2017 as the training edge, the cooperation in 2018 as the validation edge and the cooperation in 2019 as the testing edge. For graph classification task, we choose D&D and ENZYMES datasets. D&D and ENZYMES includes graphs of protein structures, in which a node represents an amino acid. We perform 10-fold cross-validation to observe model performance and reported the accuracy averaged over 10-fold.

**Backbone settings.** As for regular-scale datasets Cora, Citeseer and PubMed, we adopt GCN/GIN/-GAT backbones for node classification and link prediction tasks. In our implementation, we adopt 3-layer GCN, 3-layer GIN and 3-layer GAT, respectively. For link prediction in Ogbl-Collab, we adopt 28-layer ResGCNs. For graph classification, we hope to obtain a small size graph representation and we use GraphSAGE for forecasting. Concretely, we select three sparsification layers for graph size scaling. After the last sparsification layer, we add a GrpahSAGE layer and MLP for prediction. Further, we place the training details and hyper-parameter configuration in Table 6.

**Weight sparsity settings.** In our implementation, we control the sparsity of each individual layer to be equal to the total sparsity and random choice weights in each layer. In this setting, we keep the first layer dense, since sparsifying this layer has a disproportional effect on the performance and almost no effect on the total size.

**Hierarchical graph sparsification ratio.** For graph classification, we found that $n_l$ has little effect on the results, while the final $n_L$ can be limited to a small size (even 95% graph sparsity). As for node classification and link prediction tasks, we should control $n_l$ to be slightly larger than $n_{l+1}$ in hierarchical graph sparsification (HGS) process.

Table 6: Training details and hyper-parameter configuration. $\xi^{(0)}$ and $\rho^{(0)}$ indicate the starting value of the graph regularization and weight regularization, respectively. $\xi_a$ and $\rho_a$ indicate the size of the graph regularization and weight regularization increase value.

| | Computing Infrastructures: three NIVIDIA Tesla v100 (16GB GPU) Software Framework: Pytorch | | | | | | | | |
|---|---|---|---|---|---|---|---|---|---|
| Task | Node classification | | | Link prediction | | | | Graph classification | |
| Dataset | Cora | Citeseer | PubMed | Cora | Citeseer | PubMed | Ogbl-collab | D&D | ENZYMES |
| Epochs (pre-train/fine-tune) | 500/200 | 500/200 | 700/200 | 500/200 | 500/200 | 700/200 | 500/200 | 500/300 | 500/300 |
| $\xi^{(0)}$ | 5e-04 | 5e-04 | 5e-04 | 5e-04 | 5e-04 | 5e-04 | 5e-04 | 5e-04 | 5e-04 |
| $\xi_a$ | 1e-06 | 1e-06 | 1e-06 | 1e-06 | 1e-06 | 1e-06 | 1e-06 | 1e-06 | 1e-06 |
| $\rho^{(0)}$ | 5e-04 | 5e-04 | 5e-04 | 5e-04 | 5e-04 | 5e-04 | 5e-04 | 5e-04 | 5e-04 |
| $\rho_a$ | 1e-06 | 1e-06 | 1e-06 | 1e-06 | 1e-06 | 1e-06 | 1e-06 | 1e-06 | 1e-06 |
| Optimizer | Adam | Adam | Adam | Adam | Adam | Adam | Adam | Adam | Adam |
| learning rate | 0.008 | 0.01 | 0.01 | 0.008 | 0.01 | 0.01 | 0.01 | 0.0001 | 0.0001 |

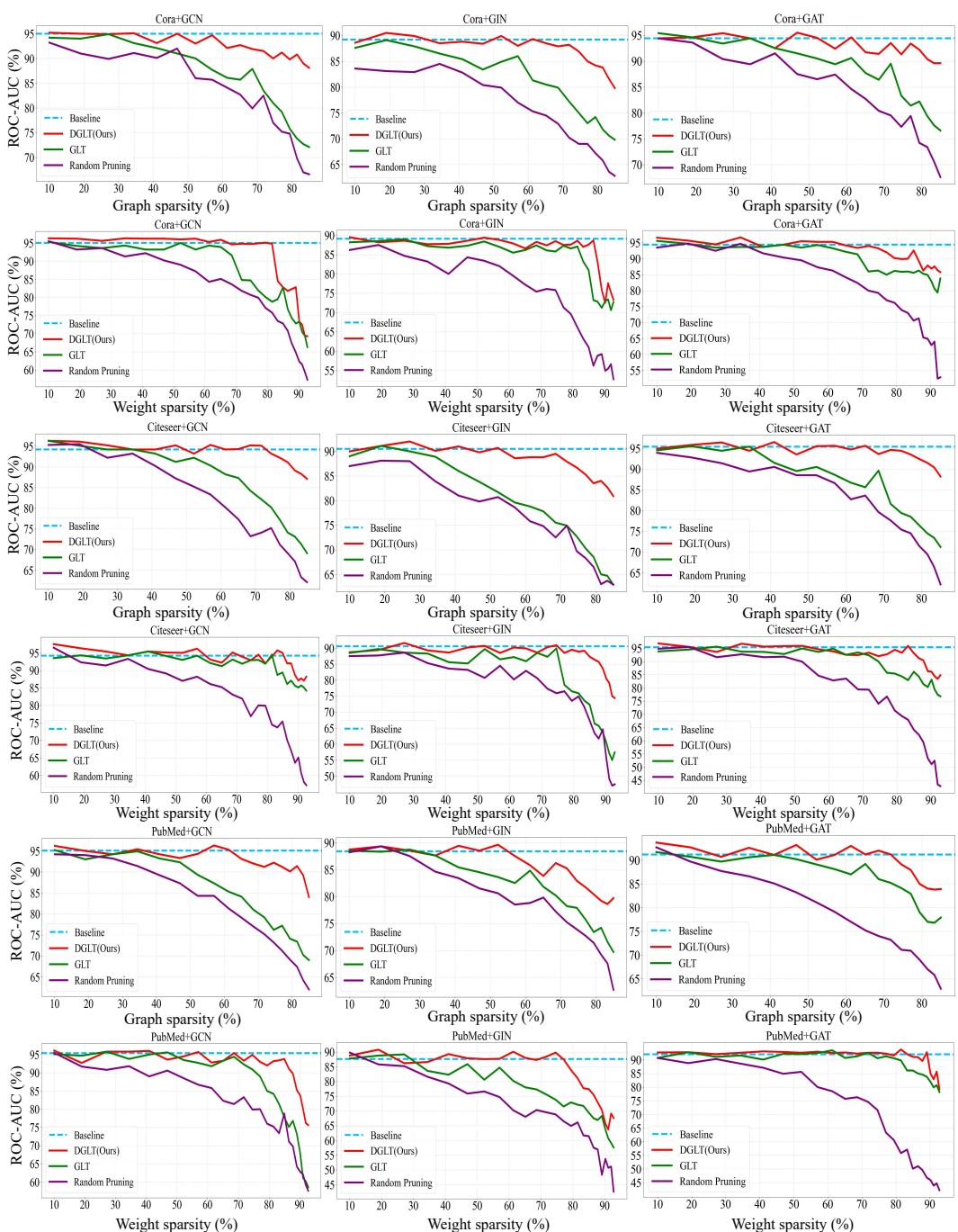

Figure 7: The results for additional **link prediction task**. We adopt Cora/Citeseer/PubMed as benchmarks and test our DGLT alogrithms on GCN/GIN/GAT three backbones for comparison. Blue dash lines represent the baseline performance of full GNNs on full graphs.

## F  ADDITIONAL EXPERIMENTS TO ANSWER RQ1

More experiments on **link prediction task** are shown in Fig. 7. We make observations as follows: (1) DGLT aggressively improves the reasoning efficiency without significant performance degradation. For Cora dataset, we can get graph lottery tickets with nearly 50% graph sparsity and 80% weight sparsity. For Citeseer dataset, we can get graph lottery tickets with nearly 70% graph sparsity and 85% weight sparsity. For PubMed dataset, we can get graph lottery tickets with nearly 60%

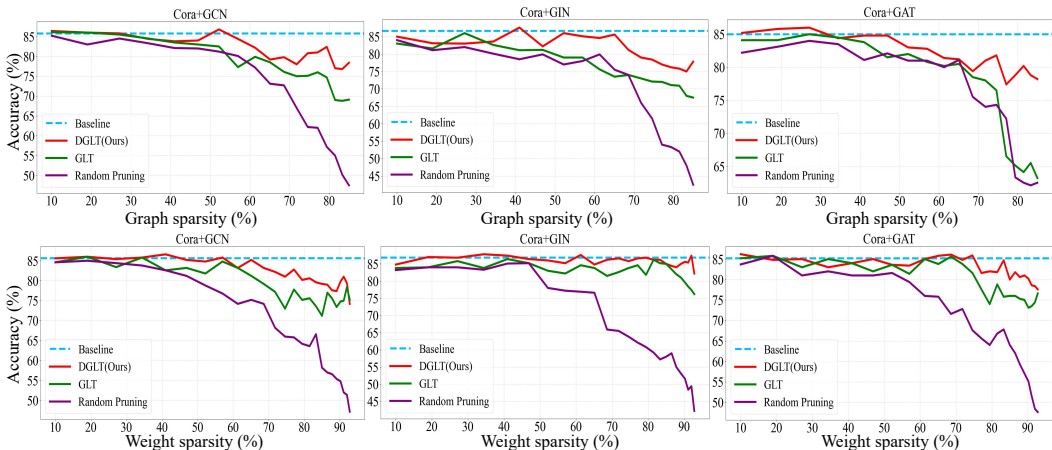

Figure 8: The results for **node classification task** on Cora benchmark. Blue dash lines represent the baseline performance of full GNNs on full graphs.

graph sparsity and 80% weight sparsity. (2) DGLT can transfer seamlessly to different graph-related tasks. In addition to the node classification task, DGLT model can also show better results on the link prediction task. (3) We can get similar conclusions as in the main experiments, DGLT can achieve better results than GLT and random pruning under the same sparsity ratio.

Further, we conduct 3-layer GraphSAGE Hamilton et al. (2017) counterpart with DGLT algorithm on ENEYMES and D&D datasets under different weight sparsity ratio. Similar to the original graph pooling method DIFFPOOL settings, we control the graph sparsity as 95% (total two pooling layers in DIFFPOOL) and observe the performance under the different weight sparsity. As shown in Table 7, it is not difficult to find that the information extrusion algorithm well maintains the accuracy under high sparsity. On ENEYMES dataset, We can observe that the expressiveness of the model could be improved even pruning 80% weights. Meanwhile, when the weight sparsity of the model reached 90%, the performance of the model still did not decrease significantly: it only decreased by 2.24% on the ENEYMES dataset and by 0.62% on the D&D dataset. In a nutshell, these results demonstrate that our DGLT algorithm can also be easily generalized to graph classification tasks, which further illustrates that DGLT has resilient pluggability.

Table 7: Graph classification accuracies in percent over ENZYMES and D&D datasets under different weight sparsity.

| DIFFPOOL+ENZYMES; Baseline accuracy:63.21% Graph sparsity=95% | | | | | | | | |
|---|---|---|---|---|---|---|---|---|
| Weight sparsity (%) | 10 | 20 | 30 | 40 | 50 | 60 | 70 | 80 | 90 |
| DGLT (Acc) | 65.27±0.33 | 65.44±0.49 | 66.53±0.37 | 63.55±0.44 | 62.49±0.47 | 62.17±0.57 | 63.21±0.54 | 61.77±0.47 | 60.97±0.53 |

| DIFFPOOL+D&D; Baseline accuracy:77.33% Graph sparsity=95% | | | | | | | | |
|---|---|---|---|---|---|---|---|---|
| Weight sparsity (%) | 10 | 20 | 30 | 40 | 50 | 60 | 70 | 80 | 90 |
| DGLT (Acc) | 80.17±0.34 | 77.38±0.47 | 76.39±0.55 | 77.670.78 | 78.50±0.61 | 77.51±0.47 | 77.36±0.47 | 78.39±0.56 | 76.71±0.77 |

# G  ADDITIONAL ABLATION RESULTS

In addition to Cora+GCN setting on Sec. 4.4, we summarize Citeseer+GAT experimental results on **link prediction task** and PubMed+GIN experimental results on **node classification task**. As shown in Fig. 9, we can further draw similar conclusions that our DGLT algorithm can obtain graph lottery tickets, which can both save memory footprints and speed up reasoning efficiency.

# H  CONNECTIONS TO GNN EXPLAINABILITY

GNN explainability research line mainly focuses on identifying a key subgraph of the full graph—that is, what knowledge drives the GNN model to make a certain prediction? They cast

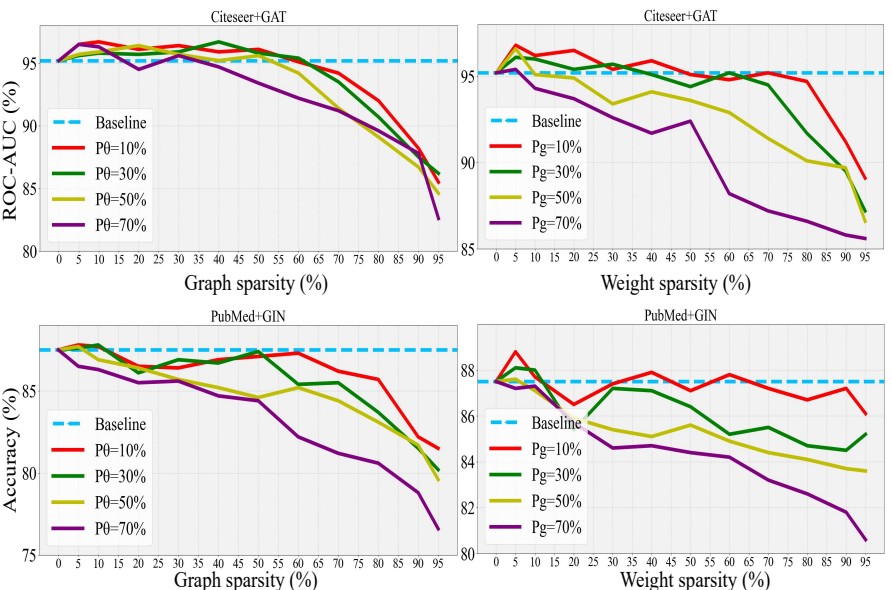

Figure 9: Ablation studies of pruning ratios for transforming sparse graph $p_g$ and network $p_\theta$, we select Citeseer+GAT for link prediction task and PubMed+GIN for node classification task. Bule dash lines represent model performance over full graph counterpart with full networks.

the learning paradigm of GNN as minimizing the empirical risk with the masked subgraphs, which are regarded as rationales to guide the model predictions Chang et al. (2020); Miao et al. (2022); Creager et al. (2021); Wu et al. (2022b); Sui et al. (2022). Our explanation focuses on why the transformed subgraph can make reliable predictions, which is similar to those described above. In our implementation, we want to give an explanation from a graph information bottleneck perspective about why the transformed subgraph has expressive ability. This "explanations" process (Section 3.3) is similar to Miao et al. (2022), the difference is that our subgraph is not a fraction of the original full graph but is obtained by a hierarchical graph sparsification (HGS) algorithm. However, to our best knowledge, this is the first work to transform a graph lottery ticket from the information bottleneck perspective and we will explore how the relationships between GNN explainability with gradually increased regularization in the future work.

## I    DISCUSSION OF DGLT

In this work, we follow the perspective of the Dual Lottery Ticket Hypothesis (DLTH) and investigate GNN subnetwork training and subgraph identifying from a complementary direction—that is, given a specific substructure of GNN model or size of adjacency matrix, we can always transform them to a winning graph lottery ticket (please note, we consider the common case based on uniformly random selection for GNN model or subgraph compression form, not including certain extreme situations such as the disconnected subnetworks or subgraph). This conjecture, if it is true, has rather promising practical implications—it may suggests that the message passing function (i.e., information aggregation) of training a GNN model is in fact unnecessary as one only needs to select a target size of adjacency matrix or target substructure of GNN, and then use hierarchical graph sparsification (HGS) algorithm or gradually increased regularization for information extrusion.

**Comparisons with DLTH**. DLTH focuses on transforming a randomly initialized dense network into an admirable subnetwork, which can achieve better at least comparable to LTH. Building upon DLTH, our DGLT firstly generalizes this idea to GNN, and investigates a more universal yet challenging problem—that is, how to transform a randomly selected ticket (i.e., a pair of graph and network) to a graph lottery ticket in GNNs? However, the key tool of DLTH—gradually increased regularization—is not applicable to graphs due to the fixed structure. To this end, we adopt HGS to break this gap and adjustably pre-define the substructure of adjacency matrices. Compared with the

infeasibility of DLTH in graph compression, our method makes it possible to sparse the graph and GNN targeting to the specified structure.

