# OpenReview forum: "Searching Lottery Tickets in Graph Neural Networks: A Dual Perspective"
_ICLR.cc/2023/Conference — ICLR 2023 poster_

### Official Review · Reviewer_yVH3 · 2022-10-24

**Confidence:** 3
**Correctness:** 3
**Technical Novelty And Significance:** 2
**Empirical Novelty And Significance:** 3
**Recommendation:** 6

**Clarity, Quality, Novelty And Reproducibility:**

In general, this paper looks novel to me (with the first attempt to apply DLTH onto graphs and GNNs), with good clarity and a nice presentation. I found this paper interesting but not that exciting, mainly because of the way the graph lottery ticket is obtained (please refer to my first weaknesses). Some other concerns are over the experimental settings and results.

**Strength And Weaknesses:**

Strengths:
1.	As far as I know, this paper delivered the first attempt to generalize the Dual Lottery Ticket Hypothesis onto graph data and GNNs to obtain the graph lottery ticket.
2.	Empirical study is clean and shows promising results on some datasets.
3.	This paper is well-written with a nice presentation. Many sections are easy to follow.

Weaknesses:
1.	As far as I understand, the finally found dual graph lottery ticket is obtained after the training of the GNNs, i.e., the ticket contains fully trained parameters. This makes me feel the method proposed in this paper is more like a graph/model pruning method where we simultaneously train and prune a GNN until we finally find a sparse subgraph/subnetwork with non-dropped accuracy. If so, the dual perspective of the lottery ticket hypothesis does not that novel or interesting to me. I would like to see more explanations or justifications from the authors on this point.
2.	If my understanding above is true, the authors are also encouraged to compare their method with standard compression-based methods.
3.	As stated in the introduction, the Graph Lottery Ticket hypothesis can simultaneously simplify the input graph and prune the GNNs without compromising model performance. However, there exists an unignorable gap between the performance of the pruned GNNs with a pruning ratio above ~60% or 70% and the full-graph baselines on Citeseer and Ogbl-Collab. Is there any possible explanation for this phenomenon?

Minor concerns:
1.	All experiment results in the current manuscript do not have standard deviations. Authors are strongly encouraged to add them by multiple random runs since on several datasets the margin between some methods is very close. Also, single-run-based results could sometimes be tricky due to e.g., cherry-picking.
2.	Since this paper also tackles the compression of input graphs, I believe it is more meaningful to include “truly” large-scale graphs in the experiments. I believe nowadays large-scale graphs are typically at least in the scale of the millions in terms of the number of nodes, e.g., OGB-Products and ogbl-citation2.
3.	Missing references to some (potentially) very related paper: Inductive Lottery Ticket Learning for Graph Neural Networks, 2022.


**Summary Of The Paper:**

This paper considered scaling up Graph Neural Networks (GNNs) via the input graph and model (GNNs) compression while maintaining model performance. The authors considered a dual problem of the original Graph Lottery Ticket hypothesis problem, i.e., transforming a random pair of input graphs and networks into a graph lottery ticket. The proposed framework called the Dual Graph Lottery Ticket (DGLT), utilizes the $L_2$ regularization to achieve sparsity and a hierarchical graph representation learning procedure to learn the sparse subgraph. In addition, a one-shot pre-train and finetune step are applied to prune the GNNs. The authors also provided some theoretical analyses to show that their proposed DGLT could achieve better subgraphs compared with sampling-based methods. Experiment results are provided on the proposed method’s effectiveness, scalability on large graphs and ablation study, etc.

**Summary Of The Review:**

Overall, I found this paper interesting in exploring DLTH under the GNN setting. However, there are some concerns over the technical novelty as well as some empirical results. My final scores will be largely upon the discussion with other reviewers. I am open to hearing from the authors and willing to change my scores if I made any misunderstandings.

---

> ### Author Response · Authors · 2022-11-11
> **Response to reviewer yVH3**
>
> Thank you very much for spending a huge amount of time on our paper, and we are really encouraged to hear your recognition of our work!   Based on your questions and recommendations, we have made many revisions that significantly improved the clarity of the paper.  Your effort really makes our paper solid!
>
> > **Comment 1.  (1) The method proposed in this paper is more like a graph/model pruning method where we simultaneously train and prune a GNN until we finally find a sparse subgraph/subnetwork with non-dropped accuracy. (2) There exists an unignorable gap between the performance of the pruned GNNs with a pruning ratio above ~60% or 70% and the full-graph baselines on Citeseer and Ogbl-Collab. Is there any possible explanation for this phenomenon?**
>
> (1) Key idea of the DGLT:
> + DGLT follows the idea of dual lottery tickets training paradigm (i.e., pretrain-prune-retrain) and tries to explore a more challenging and general case given a specific substructure of the GNN model or size of adjacency matrix, we can always transform them to a winning graph lottery ticket—which is a dual perspective of GLT algorithm.
>
> (2) Difference between GNN pruning/compression algorithms:
> + Different from the traditional GNN pruning/compression algorithms, our DGLT first points out that transforming a graph into a predefined smaller size can lead to no performance degradation. Compared with graph prunning (like GLT) or graph compression (Graph Condensation [1]), DGLT claims that a randomly predefined graph can be transformed into a proper condition with a highly-informative form. This conjecture, if it is true, has rather promising practical implications—it suggests that the message passing function (i.e., information aggregation) of training a GNN model is in fact unnecessary as one only needs to select a target size of adjacency matrix or target substructure of GNN, and then use hierarchical graph sparsification (HGS) algorithm or gradually increased regularization for information extrusion.
>
> Our major contribution and novelty lie in uncovering the relationships between full graphs and arbitrary form subgraphs, but not **concentrate solely on the training and pruning a GNN until we finally find a sparse subgraph/subnetwork without sacrificing accuracy**.
>
>
> (3) These gaps depend on the property of the graph datasets, such as feature dimensionality and the dense connections in graphs.
>
> + The unignorable gaps between full-graph baselines and above ~60% or 70% pruning ratio on Citeseer may result from the higher node feature dimensions (3703-d). For example, a very high dimension may complicate the information extrusion to the target substructure. Conversely, we argue that the gaps in the Ogbl-collab dataset result from the dense connections between nodes. In practice, it is extremely hard to compress such dense connections of a large graph into a small subgraph without performance dropping.
>
> > **Minor concerns: 1. All experiment results in the current manuscript do not have standard deviations.  2. Including “truly” large-scale graph datasets in the experiments.  3. Missing references to some (potentially) very related paper: Inductive Lottery Ticket Learning for Graph Neural Networks, 2022.**
>
> + Thank you for bringing this to our attention. These reviews really help us strengthen our results! We agree that it's extremely relevant (Inductive Lottery Ticket Learning for Graph Neural Networks, 2022.) to our work, and we have added a citation to this work in the revised version. Meanwhile, we have run each experiment three times and reported their means with standard deviations, and we **have carefully organized these results into revisions (Tab 3 and Tab 7) for facilitating your reading!**
>
> + We have tested our DGLT on 28-layer ResGCN (Max aggregator) with ogbn-products (on NVIDIA Tesla V100 32GB GPU), here we follow the experiment setting with original paper [2] and control pruning ratio of weight equal to 20% for comparison. From our reports, we can easily find that DGLT outperforms GLT and can obtain a better graph winning ticket. we will try to report more results before the end of the rebuttal:
>
>
> | Graph sparsity      | 0\%         | 30\%       |  50\%    | 60\%   |   70\%  |      80\%    |
> | ----- | ------------- | --------------- | --------------- | --------------- |--------------- |--------------- |
> | ResGCN+GLT| 79.83 $\pm$ 0.35 | 78.44 $\pm$ 0.44 |77.91 $\pm$ 0.51 | 75.56 $\pm$ 0.62 | 72.82 $\pm$ 0.67 | 63.88 $\pm$ 1.35 |
> | ResGCN+DGLT   | 80.36 $\pm$ 0.25 | 79.45 $\pm$ 0.38 | 78.30 $\pm$ 0.47 | 77.21 $\pm$ 0.59 | 73.52 $\pm$ 0.75 | 64.73 $\pm$ 1.47 |
>
>
> Thank you again for the valuable feedback. Please let us know if you have any other comments/questions.
>
> Best,
>
> Authors.
>
>
> Reference:
>
> [1] Graph condensation for graph neural networks[J]. arXiv preprint arXiv:2110.07580, 2021.
>
> [2] DeeperGCN: All You Need to Train Deeper GCNs

---

### Official Review · Reviewer_PXxs · 2022-10-25

**Confidence:** 3
**Correctness:** 3
**Technical Novelty And Significance:** 3
**Empirical Novelty And Significance:** 3
**Recommendation:** 6

**Clarity, Quality, Novelty And Reproducibility:**

The motivation and organization are clear but lack some technique detail. Adding more graph property related designs would be better to improve the novelty.

**Strength And Weaknesses:**

Strength:
* It is interesting to extend the dual lottery tickets hypothesis into graph learning and jointly get well-performed sparse subgraphs and subnetworks from randomly selected ones.
* The design of hierarchical graph sparsification makes it possible to perform graph classification tasks.

Weakness:
* The idea of the dual lottery ticket is to prove the universality of tickets in neural networks. However, the design of DGLT does not contain a ‘random graph’. The authors haven’t mentioned how to predefine the subgraph structure, instead, they only constrain the number of nodes in this graph.
* The notion of hierarchical graph sparsification may be misleading. Graph sparsification means nodes with fewer connections rather than reducing the number of nodes in one graph with the hierarchical graph strategy. Besides, how to evaluate prediction for the hierarchical graph is vague because the label for nodes in the hierarchical graph is unknown.
* The strategy of gradually increased regularization has already been proposed in [1]. It would be better if the authors could provide more differences for this strategy between DGLT and DLT.
* Some technical detail needs further illustration. For example, how to uniformly sample subnetworks in DGLT should be described in more information. Meanwhile, good explainability is expected to have some empirical results to support it along with the GIB-based analysis.

[1] Dual Lottery Ticket Hypothesis, ICLR 2022



**Summary Of The Paper:**

This paper proposes the Dual Graph Lottery Ticket framework to obtain the triple-win graph lottery ticket. Regularization-based network pruning and hierarchical graph sparsification are designed to jointly get the sparsified graph and subnetwork. What’s more, the graph information theory guarantees the explainability of this triple-win lottery ticket.

**Summary Of The Review:**

It’s really interesting to transform a random ticket into a graph lottery one and the authors introduce gradually increased regularization terms and hierarchical graph sparsification to achieve this goal. Besides, the Graph Information Bottleneck theory is utilized to guarantee a more explainable ticket. However, the design of the random ticket and the technical detail to find DGLT need to be further illustrated and the explainability of DGLT needs more experiments to support it.

---

> ### Author Response · Authors · 2022-11-10
> **Response to reviewer PXxs**
>
> Thank you for your thoughtful and constructive reviews of our manuscript. Based on your questions and recommendations, we have made many revisions that significantly improved the clarity of the paper. Below, we give point-by-point responses to your comments and describe the revisions we made to address them.
>
> > **Comment 1. The idea of the dual lottery ticket is to prove the universality of tickets in neural networks. However, the design of DGLT does not contain a ‘random graph’. The authors haven’t mentioned how to predefine the subgraph structure, instead, they only constrain the number of nodes in this graph.**
>
> Dual lottery ticket articulates that: *A randomly selected subnetwork from a randomly initialized dense network can be transformed into a trainable condition, where the transformed subnetwork can be trained in isolation and achieve better at least comparable performance to LTH.*
>
> We generalize this idea to graph and GNN model, however, it is impossible to achieve randomness in the graph, since its adjacency matrix $A$ and features are given when the graph is input. In graph lottery ticket (GLT) hypothesis [1], the UGS algorithm adopts a randomly initialized mask to attach to the adjacency matrix for edge selection.  We consider that the idea of “randomly initialized” only exists in trainable weights and we believe that our “randomness in graph” can be understood from two perspectives:
>
> + Random soft mask for adjacency matrix from scratch: we train a randomly initialized soft mask on $A$ from the scratch, and finally send the combination of ${m}_{A}^{(L)} \bigodot A^{(L)}$ to predictor for prediction to predictor for prediction.
> + Random size of compressed $A$: In our implementation, we consider a more general case—that is, transform a graph to a pre-defined target size but not recklessly remove the edges. **We claim that this is the first work explored to transform a fixed size graph to a randomly pre-defined size.**
>
> [1] A Unified Lottery Ticket Hypothesis for Graph Neural Networks. ICML 21
>
> > **The notion of hierarchical graph sparsification may be misleading**
>
> Sorry for this confusion. Graph sparsification can be concluded into edge-removal models [2][3], node-removal models [4][5], and some compression graph algorithms [6][7]. Our work can be deemed as a compression graph algorithm, which compress original graph dataset into a small, synthetic and highly-informative graph but not node-removal algorithm. **We'd be happy to add a section “Related work on Graph Sparsification” on Appendix if you think it's necessary.**
>
> Different from GLT, when we compress graph data into small size, we add a GNN+MLP for dimension mapping. For graph classification, we map graph into 1-d label (similar to readout layer). For node- or edge- wise prediction, we have control MLP to achieve dimension match. This is similar to currently popular graph condensation [7].
>
> [2] A unified lottery ticket hypothesis for graph neural networks ICML 21
>
> [3] Spielman D A, Teng S H. Spectral sparsification of graphs[J]. SIAM Journal on Computing, 2011, 40(4): 981-1025.
>
> [4] Spectrally approximating large graphs with smaller graphs ICML 18
>
> [5] Loukas A. Graph Reduction with Spectral and Cut Guarantees[J]. J. Mach. Learn. Res., 2019, 20(116): 1-42.
>
> [6] Cluster-gcn: An efficient algorithm for training deep and large graph convolutional networks  Proceedings of the 25th ACM SIGKDD
>
> [7] Graph condensation for graph neural networks[J]. arXiv preprint arXiv:2110.07580, 2021.
>
>
> > **Comment 3. It would be better if the authors could provide more differences for this strategy between DGLT and DLT.**
>
> We agree! We have already added a section “Discussion of DGLT” in **Appendix I** to provide more differences for this strategy between DGLT and DLT.
>
> > **Comment 4. Some technical detail needs further illustration and good explainability is expected to have some empirical results to support it along with the GIB-based analysis.**
>
> We agree! In our implementation, we control the sparsity of each individual layer to be equal to the total sparsity and random choice weights in each layer (Pytorch np.random.choice function). In this setting, we keep the first layer dense, since sparsifying this layer has a disproportional effect on the performance and almost no effect on the total size. Follow your advice, we have added descriptions in **Appendix E (Experimental Settings)**.
>
> As for good explainability, in our work, we proposed the HGS algorithm which builds upon GIB theory, all graph sparsity-related experiments (of GLT and DGLT) can prove that the information extrusion methods are better than the sampling-based methods. Despite this, gradually increased regularization has already existed in some practical usage [8][9], our GIB theory goes from the theoretical point of view to support these practical usages.
>
> [8] Neural pruning via growing regularization. ICLR 21
>
> [9] Dual lottery ticket hypothesis. ICLR 22

---

> > ### Comment · Reviewer_PXxs · 2022-11-27
> > **Acknowledgement and Thank You for Your Response**
> >
> > Many thanks for your thoughtful and thorough feedback, especially on the explanation of the randomness in graphs and the added discussion on the graph sparsification of related works, which are helpful. Meanwhile, I have also read the authors' responses to other review comments, which helps me form a better understanding of this paper.
> >
> > I am happy to increase my score. Hopefully, the authors could include these points as well as the discussion in the updated version.

---

> > > ### Author Response · Authors · 2022-11-28
> > > **Thank you for raising the score:**
> > >
> > > Thank you for your careful and detailed reading of our paper and responses! We are working on incorporating reviewer comments into an updated version of the draft. Your expertise significantly helps us strengthen our manuscript!

---

### Official Review · Reviewer_rozM · 2022-10-27

**Confidence:** 3
**Correctness:** 3
**Technical Novelty And Significance:** 4
**Empirical Novelty And Significance:** 4
**Recommendation:** 8

**Clarity, Quality, Novelty And Reproducibility:**

Overall, the paper is well-written and most of the different concepts are clearly presented. At the same time, the paper keeps a good balance between theoretical contributions and empirical evaluation. Regarding the methodology, I believe the paper makes a novel contribution to the field.

**Strength And Weaknesses:**

**Strengths:**
* The paper presents a rigorous formulation of the GLT problem on graphs. Efforts have also been made to theoretically explain the performance of the model.

* Detailed empirical analysis, studying different aspects of the methodology.

**Weaknesses:**
* A point that is not very clear in the paper has to do with the practical utility of the theoretical observations that are made in Sec. 3.3. I cannot really see how these observations have a direct impact on the performance of the model in practice. It would be helpful if the authors could further clarify this point.

* Another point has to do with the ablation study. A core component of the methodology is based on the hierarchical graph sparsification module which is implemented as a trainable pooling operator. Oftentimes, especially on graphs with no clear modular structure, the benefit of hierarchical pooling is not always clear. The impact of this component on the performance of the model should be further analyzed.


**Summary Of The Paper:**

The paper studies a new formulation of the LTH problem on graphs. Instead of searching for a sparse subnetwork with good performance, it poses the question of how to transform a randomly selected ticket into a lottery one. Although this problem has been studied before in DNNs, the paper proposes a solution for GNNs. Specifically, the proposed methodology, called DGLT,  comprises a hierarchical graph sparsification module based on hierarchical graph pooling and a training module following a suitable adjusted L2 regularization component. In addition, the paper borrows ideas from the Information Bottleneck theory to provide performance guarantees. The model is empirically evaluated on the task of node classification.

**Summary Of The Review:**

The paper makes an interesting contribution toward further understanding graph structure simplification and model compression in GNNs using lottery tickets. Nevertheless, I still have some concerns about the theoretical analysis derived in the paper.

---

> ### Author Response · Authors · 2022-11-10
> **Response to rozM**
>
> We sincerely thank you for your careful comments and thorough understanding of our paper. Here we give point-by-point responses to your comments and describe the revisions we made to address them. We would be happy to add additional clarifications and revisions to the paper to address any additional recommendations from the reviewer.
>
> > **Comment 1. A point that is not very clear in the paper has to do with the practical utility of the theoretical observations that are made in Sec. 3.3. It would be helpful if the authors could further clarify this point.**
>
> Thank you for this constructive and thoughtful feedback. We are really encouraged to hear your recognition of our work!
> In this work, we follow the perspective of the Dual Lottery Ticket Hypothesis (DLTH) [1] and investigate GNN subnetwork training and subgraph identifying from a complementary direction—that is, given a specific substructure of GNN model or size of adjacency matrix, we can always transform them to a winning graph lottery ticket (please note, we consider the common case based on uniformly random selection for GNN model or subgraph compression form, not including certain extreme situations such as the disconnected subnetworks or subgraph).
>
> This conjecture, if it is true, has rather promising practical implications—it suggests that the message passing function (i.e., information aggregation) of training a GNN model is indeed unnecessary as one only needs to select a target size of adjacency matrix or target substructure of GNN, and then use hierarchical graph sparsification (HGS) algorithm or gradually increased regularization for information extrusion.
>
> We agree that “it would be helpful if the authors could further clarify this point” and we have added **Discussion of DGLT** in Appendix I. If you have further questions, please feel free to contact us.
>
>
> > **Comment 2. Another point has to do with the ablation study. A core component of the methodology is based on the hierarchical graph sparsification module which is implemented as a trainable pooling operator. Oftentimes, especially on graphs with no clear modular structure, the benefit of hierarchical pooling is not always clear. The impact of this component on the performance of the model should be further analyzed.**
>
> Thanks again for making our results even stronger! This is really an interesting issue. For graph classification, we found that $n_{l}$ has little effect on the results, while the final $n_{L}$ can be limited to a small size (even 95% graph sparsity). As for node classification and link prediction tasks, we should control n_l to be slightly larger than $n_{l+1}$ in hierarchical graph sparsification (HGS) process.
>
> However, with the change of $n_{l}$, the parameters of the entire network are also changing, we cannot accurately explore the influence of a single variable $n_{l}$ on the model, but we have mentioned problems that should be paid attention to in view of the different sparsity of different tasks and the issue of hierarchical changes of $n_{l}$ in the experiments. **Follow you advice, we have added an additional description in Appendix E.**
>
> Thank you again for the valuable feedback. Please let us know if you have any other comments/questions.
>
> Best regards,
>
> Authors
>
>
>
> Reference:
>
> [1] Dual lottery ticket hypothesis[J]. arXiv preprint arXiv:2203.04248, 2022.

---

### Official Review · Reviewer_xzm4 · 2022-11-02

**Confidence:** 4
**Correctness:** 3
**Technical Novelty And Significance:** 2
**Empirical Novelty And Significance:** 3
**Recommendation:** 6

**Clarity, Quality, Novelty And Reproducibility:**

The clarity and quality of the paper presentation is generally satisfactory, and can be improved by providing more details/justifications about the questions I raised in the summary. The novelty and technical contribution is limited, as the dual lottery ticket hypothesis and pooling-based sparsification are not newly proposed in this work, but it is still good to find that such a combination provides empirical improvement. The codes have been included for reproducibility.

**Strength And Weaknesses:**

**Strength**
- The dual lottery ticket hypothesis is firstly studied on graph lottery ticket.
- The graph pooling technique is introduced to replace the sampling-based graph sparsification.
- Extensive experiments are conducted for multiple applications.

**Weaknesses**
- Important details about the training loss are not provided, especially when the dimension of the sparsified graph is changed by pooling operations.
- Some claims made by the paper are not well justified, e.g. better explainability, pooling is better than sampling-based algorithms.
- The theoretical analysis of the proposed method is not clear enough.


**Summary Of The Paper:**

This paper explores the dual lottery ticket hypothesis on graphs. Specifically, a pair of core subgraph and sparse subnetwork is jointly uncovered via regularization-based network pruning and hierarchical graph sparsification. The method is also analyzed from the perspective of graph information bottleneck. Experiments on node classification, graph classification and link prediction tasks show the effectiveness of the proposed method compared with GLT.

**Summary Of The Review:**

The paper introduces dual lottery ticket hypothesis and graph pooling in graph lottery ticket. In general the paper is well written with clear motivation and extensive experiments are conducted. However, I have concerns about the limited technical contribution and unclear details.

=== Method ===

1. The example in Figure 1 is not convincing. Aggregation failure may also happen in the proposed sparsification, since mask matrices are also applied to the adjacency matrix (e.g. $m_A$) which removes edges among clusters/communities.
2. The reason for imposing a mask $m^{(l)}_A$ in each layer is not well justified. If the graph is already pooled, why does it need to be further sparsified? What is the performance if removing the masks $m_A$?

3. The calculation of the cross-entropy loss in Eq. (4) is crucial to understand how the method can serve for different tasks, but is not explained at all. If this is a node classification task, the label is per-node. However, after pooling and sparsification, the final subgraph $A^{(L)}$ and $X^{(L)}$ is no longer node-wise but cluster-wise (e.g. the dimension of $A^{(L)}$ is $n_{L}\times n_{L}$). There is clearly a gap between node-wise labels and cluster-wise embeddings. What is the specific form of the cross-entropy loss for node classification, link prediction and graph classification task, respectively?

4. The claim of better explainability is not demonstrated. It usually has a specific meaning of providing explanations to the model predictions.

5. Is $G_{sub}$ the same as $G_{hgs}$?

6. The theoretical analysis is a bit vague, especially when it comes to the proposed method. For example, the logic of why the training mask will make $I(\hat{\mathcal{G}}^*_s, Y)>I(\mathcal{G}^*_s, Y)$ is unclear. The analysis is more like intuition instead of rigorous theoretical proof. Meanwhile, Figure 3 is hard to understand. After reading the analysis, it is still unclear to me why DGLT can be comparable/better than sampling-based algorithms.

=== Experiment ===

7. What is the definition of weight sparsity and graph sparsity in the experiment?

8. How sensitive is the method to the pooling-related hyperparameters, e.g. $n_{l}$?

=== Typos ===

There are also some typos. To name a few:
- In the paragraph after Eq. (2), “assignment matrix S^{(l)}... can projecting coarsened subgraph xxx” -> “Assignment … can project …”
- Duplicate “and” in the paragraph before Lemma 1.
- “a admirable” -> “an admirable” in the paragraph before Lemma 1.

---

> ### Author Response · Authors · 2022-11-10
> **Response to reviewer xzm4 (part 1)**
>
> Thank you very much for spending a huge amount of time on our paper. Your thoughtful and constructive reviews make our results even stronger! We give point-by-point responses to your comments and describe the revisions we made to address them:
>
> > **Comment 1.  Aggregation failure may also happen in the proposed sparsification.**
>
> We are sorry for this confusing statement. Our hierarchical graph sparsification (HGS) algorithm follows a pretrain-prune-retrain process, and we demonstrate that:
> (1) Pretrain stage: We attach a **soft** mask $m_{A}^{(l)}$ element-wise product with $A^{(l)}$ at each GNN output layer, different from the hard mask (only has 0 or 1 elements in a mask),  $m_{A}^{(l)}\bigodot A^{(l)}$ is still a dense matrix. We conduct GNN to learn an assignment matrix $S^{(l)}$ for compressing $A^{(l)}$ and $X^{(l)}$ to formalize a small adjacency matrix $A^{(l+1)}$ and input feature $X^{(l+1)}$ representations. The compose of $A^{(l+1)}$ is not 0 or 1 but is a dense number. This clustering-based method enables us to effectively aggregate information from different clusters/communities with less information loss [1].
>
> (2) Retrain stage: Due to increased regularization, some elements of $m_{A}^{(l)}$ tend to be 0 and the others are still dense. Although $m_{A}^{(l)}\bigodot A^{(l)}$ tend to be sparse, we adopt hierarchical clustering, $A^{(l + 1)} = {S^{(l)}}^{T}A^{(l)}S^{(l)}$, this matrix multiplication enables $A^{(l+1)}$ is dense enough.
>
> [1] Structural entropy guided graph hierarchical pooling[C]//International Conference on Machine Learning. PMLR, 2022: 24017-24030.
>
> > **Comment 2. The reason for imposing a mask $m_{A}^{(l)}$ in each layer should justified and what is the performance if removing the masks $m_{A}$.**
>
> Sorry that we don't make it clear. In our work, we generalize the recently popular *dual lottery tickets hypothesis (DLTH)*[2] to GNNs for the first time. We not only study a more challenging and general case of investigating the internal relationships between a full GNN model and its sparse counterpart and achieving promising performances for sparse network training but also **uncover the relationships between the full graph and its predefined compression forms for specific structures**. We try to explore how expressive ability of counterpart
> $m_{A}^{(l)}\bigodot A^{(l)}$ when imposing a mask $m_{A}^{(l)}$ in each layer to discard some elements of $A^{(l)}$, and what the model performs under this circumstance. To our best knowledge, this is the first work to consider this challenging issue.
>
> In Section 4.2 (RQ1), we investigate how accuracy changes with the growth of graph sparsity, we fix the weight sparsity to zero for stability, and vice versa. When fixing graph sparsity to zero, we can consider that we have already removed $m_{A}$. As shown in Fig. 4, Fig. 7 (node classification) and Fig. 8 (link prediction), when removing the masks $m_{A}$, we can still obtain an admirable graph lottery ticket. Furthermore, DGLT consistently outperforms GLT and random pruning under the same graph/weight sparsity over all datasets, verifying the superiority of transforming a random ticket to a graph lottery ticket via information extrusion and HGS.
>
> > **Comment 3.  The calculation of the cross-entropy loss in Eq. (4) is crucial to understand how the method can serve for different tasks but is not explained at all.**
>
> Sorry that we don’t make it clear. Our work can naturally adapt to the graph classification tasks as a clustering-based graph model [2-4]. As shown in Fig. 2, we can finally obtain $m_{A}^{(l)}\bigodot A^{(l)}$ and send this to a GNN and MLP predictor for prediction (dimension mapping). As for graph classification task, the final labels are $y_{i} \in \mathbb{R}^{\mathcal{P} \times 1}$, $P$ is the number of annotated labels. The original loss function can be written as: $\mathcal{L}\left( {\mathcal{G},\Theta} \right) = - {\sum\limits_{\mathcal{P}}y_{i}}log\left( z_{i} \right)$, $z_{i} \in \mathbb{R}^{\mathcal{P} \times 1}$ is predicted labels.
>
> As for node classification and link prediction, we can obtain $m_{A}^{(L)}\bigodot A^{(L)} \in \mathbb{R}^{n_{L} \times n_{L}}$ after two non-shared GNNs. Here we send $m_{A}^{(l)}\bigodot A^{(l)}$ to a predictor (GNN+MLP) for dimension mapping (compare with GLT, we add a new predictor for matching dimension). For node classification task, assume final label $y_{i} \in \mathbb{R}^{\mathcal{J} \times 1}$, $\mathcal J$ is the total number of annotated node labels. The cross-entropy function can be written as $\mathcal{L}\left( {\mathcal{G},\Theta} \right) = - {\sum\limits_{\mathcal{J}}y_{i}}log\left( z_{i} \right)$, $z_{i} \in \mathbb{R}^{\mathcal{J} \times 1}$ is predicted labels. For link prediction task, we can get similar cross-entropy loss like node classification.
>
> [2] Self-attention graph pooling ICML19
>
> [3] Graph u-nets ICML 19
>
> [4] Hierarchical graph representation learning with differentiable pooling ICLR 18

---

> ### Author Response · Authors · 2022-11-10
> **Response to reviewer xzm4 (part 2)**
>
> > **Comment 4. The claim of better explainability is not demonstrated. It usually has a specific meaning of providing explanations to the model predictions.**
>
> Thank you for pointing out this issue. Indeed, GNN explainability mainly focus on identifying a key subgraph of full graph—that is, what knowledge drives the GNN model to make certain prediction? They cast the learning paradigm of GNN as minimizing the empirical risk with the masked subgraphs, which are regarded as rationales to guide the model predictions [5][6]. Our work is **similar** to that described above, we want to give the explains from graph information bottleneck perspective about why the transformed subgraph have the expressive ability. This “explanations” process (Section 3.3) is **similar** to [6], the difference is that our subgraph is not a fraction of the original full graph but is obtained by a hierarchical graph sparsification (HGS) algorithm. We claim that since our work is the first to involve transforming a subgraph and guaranteeing its superiority, our thinking of referring to previous work indeed easy to causes doubts. To this end, we have already added **Connections to GNN Explainability** in Appendix H.
>
> [5] Discovering invariant rationales for graph neural networks[J]. arXiv preprint arXiv:2201.12872, 2022.
>
> [6] Interpretable and generalizable graph learning via stochastic attention mechanism[C] ICML 22
>
> > **Comment 5. Is $G_{sub}$ the same as $G_{hgs}$?**
>
> Yes! Since our proof (Section 3.3) can be easily transferred to the arbitrary form of GNN, here we use different notations in the proof section. Our goal is to transform $G_{sub} \left(  G_{hgs} \right) \rightarrow {\hat{G}}_{s}^{*}$.
>
> > **Comment 6. The theoretical analysis is a bit vague, especially when it comes to the proposed method.  And how to understand Fig. 3 **
>
> In this paper, we borrow the idea of the gradually increased regularization—that is, when the penalty is increased at the same pace, because of different local curvature structures, the weights respond differently—weights with larger curvature will be less moved. As such, the magnitude discrepancy among weights will be magnified as regularization grows. Ultimately, the weights will naturally separate (unimportant weights tend to be very small and can be regarded as noise) [7][8]. This proof can be found in Appendix A.
>
> In Fig. 3, sampling-based algorithms only select nodes whose external performance is to increase $I\left( {{{\rm{G}}_{sub}}{\rm{\;}}Y} \right)$. We argue that some **features** of these nodes are harmful to increase mutual information, but sampling-based models recklessly select these nodes without feature dropping. Our DGLT tries to extrude the information to a pre-defined structure (under increased reg): under the influence of increasing regularization force, the important information in the graph converges to the substructure, making the target substructure only have reliable information and the complementary part only have harmful information.
>
>
> [7] Neural Pruning Via Growing Regularization. ICLR 21
>
> [8] Dual Lottery Ticket Hypothesis
>
> > **Comment 7. What is the definition of weight sparsity and graph sparsity in the experiment?**
>
> weight sparsity ${{\cal S}_w} = \frac{{{{\left| {\left| {{{\rm{\Theta }}^*}} \right|} \right|}_0}}}{{{{\left| {\left| {\rm{\Theta }} \right|} \right|}_0}}}$, where ${{\rm{\Theta }}^*}$ represent preselected discarded parameters in all parameters ${\rm{\Theta }}$.
>
>  Graph sparsity ${{\cal S}_g} = 1 - \frac{{||m_A^{\left( L \right)}*{A^{\left( L \right)}}|{|_0}}}{{{{\left| {\left| A \right|} \right|}_0}}}$.
>
>
> > **Comment 8. How sensitive is the method to the pooling-related hyperparameter $n_{l}$**
>
> Thanks again for making our results even stronger! This is really an interesting issue. For graph classification, we found that $n_l$ has little effect on the results, while the final $n_{L}$ can be limited to a small size (even 95% graph sparsity). As for node classification and link prediction tasks, we should control $n_l$ to be slightly larger than $n_{l+1}$ in hierarchical graph sparsification (HGS) process.
>
> However, with the change of $n_l$, the parameters of the entire network are also changing, we cannot accurately explore the influence of a single variable $n_l$ on the model, but we have mentioned problems that should be paid attention to in view of the different sparsity of different tasks and the hierarchical changes of $n_{l}$ in the experiments. **Follow you advice, we have added an additional description in Appendix E.**
>
> > **Comment 9. Typos**
>
> We have re-constructed some original descriptions and added more details to clarify our representations and have fixed all minor typos!
>
>
> Thank you again for the valuable feedback. Please let us know if you have any other comments/questions.
>
> Best,
>
> Authors

---

> ### Comment · Reviewer_xzm4 · 2022-11-27
> **Acknowledgement of authors' response**
>
> I appreciate the thorough response provided by the authors, which helped to address my concerns and improve the clarity of the paper. I would raise my score. One thing left is to make sure that some of the responses are reflected in the revised paper (e.g. definition of weight/graph sparsity).

---

> > ### Author Response · Authors · 2022-11-28
> > **Response to xzm4:**
> >
> > We have put the definition of "sparsity" in the **Metrics** of **Experimental setting (Appendix E)**, and we will seriously consider all reviewers' suggestions and include them in our paper for easy reading. We are glad our revision and rebuttal have addressed your concerns and  thank you for all of the suggestions, they have been very helpful for us to improve the paper.

---

### Official Review · Reviewer_cfDA · 2022-11-03

**Confidence:** 3
**Correctness:** 3
**Technical Novelty And Significance:** 3
**Empirical Novelty And Significance:** 3
**Recommendation:** 6

**Clarity, Quality, Novelty And Reproducibility:**

This paper is well-written and easy to follow. For novelty, the authors propose a novel DGLT framework for GNN network pruning and graph sparsification. For reproducibility, the authors should provide source code to help better understand the proposed model.

**Strength And Weaknesses:**

Strongness
1. The paper is well-written, and the motivation is intuitive and clear.
2. It is interesting to introduce Lottery Ticket Hypothesis to simplify the input graph, and it is contributive to propose a Dual Graph Lottery Ticket for GNN pruning and graph sparsification.
3. Extensive experiments demonstrate the effectiveness of the framework in terms of sparsity improvement, scalability, and performance

Weakness:
1. The model complexity in terms of training time and parameter volumes is not discussed and investigated in the paper. It is necessary to discuss the complexity of DGLT especially considering the non-shared GNNs to learn the node embeddings and the assignment matrix.
2. The authors did not investigate the performance of random pruning on node classification. Whether the pruning would affect the performance of DGLT is not detailed.

**Summary Of The Paper:**

The paper generalizes the Dual Lottery Ticket Hypothesis in GNN network pruning and graph sparsification and proposes a Dual Graph Lottery Ticket framework. Their experimental results demonstrate the effectiveness of their framework.

**Summary Of The Review:**

In general, it is a good paper with a novel framework, solid technical, and significant results. It is a modest-to-high impact paper in a subarea of graph machine learning, especially graph sparsification.

---

> ### Author Response · Authors · 2022-11-10
> **Response to reviewer cfDA**
>
> Thank you so much for spending a huge amount of time on our manuscript. Your thoughtful and constructive comments really make our work much stronger! In the following parts, we provide point-by-point responses to your questions and describe the revisions we made to address them. We would be happy to add additional clarifications and revisions to the paper to address any additional recommendations from the reviewer.
>
> > **Comment 1. The model complexity (e.g., in terms of training time and parameter volumes) should be discussed.**
>
> Thank you for raising this problem. We agree with you that the complexity should be delineated. Due to different machine GPU occupancy rates, the training time of our algorithm is extremely difficult to measure. Therefore, we use a more general form, i.e., computational complexity to illustrate the model efficiency.
>
> Following the GLT [1], we present the complexity of the DGLT algorithm. The inference time complexity of DGLT is ${\cal O} \left({{{\left| {\left| {{m_A} \bigodot {A_{all}}} \right|} \right|}_0} \times F} +  {\left| {\left| {{m^*}} \right|} \right|_0} \times \left| {\cal V} \right| \times {F^2} \right) + {\cal O}\left( {\cal K} \right)$, where ${m_A}$ represents all mask set for all adjacency matrix outputs; $F$ is the dimension of the feature; $\left| {\cal V} \right|$ is the number of nodes; ${\left| {\left| {{m^*}} \right|} \right|_0}$ represents all remained parameters of two non-shared GNNs; ${\cal O}\left( {\cal K} \right)$  represents inference time complexity of learning the node embeddings and the assignment matrix. They are obtained by multiplying multiple matrices and the inference time complexity of ${\cal O}\left( {\cal K} \right) = {\cal O}\left( {L \times {{\left| {\cal V} \right|}^3} + L \times \left| {\cal V} \right| \times F} \right)$.
>
> As for GLT, the inference time complexity of GLTs is ${\cal O}\left( {L \times {{\left| {\left| {{m_g} \bigodot A} \right|} \right|}_0} \times F + L \times {{\left| {\left| {{m_\theta }} \right|} \right|}_0} \times \left| {\cal V} \right| \times {F^2}} \right)$, where $L$ is number of layers, ${\left| {\left| {{m_g} \bigodot A} \right|} \right|_0}$ is the number of remaining edges in the sparse graph, $F$ is the dimension of the feature and $\left| {\cal V} \right|$ is the number of nodes.
>
> Since the number of sparsification in our HGS is small, the main additional parameters of DGLT (compared with GLT) are concentrated on the predictor (GNN+MLP) spliced later (Compared with GLT, we do not attach masks on parameters, which can also reduce the parameter volume).
>
> > **Comment 2. The authors did not investigate the performance of random pruning on node classification. Whether the pruning would affect the performance of DGLT is not detailed.**
>
> We apologize for making this part confusing. Indeed, GLT[1] has already investigated the performance of random pruning on node classification tasks. In our work, we have also tested the performance of random pruning on node classification on Cora, Citeseer and PubMed (as shown in Fig 4 and Fig 7). Meanwhile, for node classification, link prediction and graph classification, we have tested the DGLT performance under the different pruning ratio and described the effect of pruning on DGLT (see Sec 4.2).
>
> Hope our explanation would facilitate a better understanding. In summary, we have followed your advice to add **Complexity analysis of DGLT and GLT (Appendix D)** and **descriptions of random pruning on node classification task (Section 4.2 and Appendix F)**. Thank you again for your thoughtful comments that greatly strengthen our paper.
>
> Reference:
>
> [1] A unified lottery ticket hypothesis for graph neural networks[C]//International Conference on Machine Learning. PMLR, 2021: 1695-1706.

---

> > ### Comment · Reviewer_cfDA · 2022-11-27
> > **Thanks for your response**
> >
> > I would to thank the authors for taking the time to respond to my comments and making changes based on our suggestions. The response has addressed my concerns, and I would vote to accept the paper.

---

> > > ### Author Response · Authors · 2022-11-28
> > > **Thank you!**
> > >
> > > We thank the reviewer cfDA for more strongly supporting our paper! We are glad our revision and rebuttal have addressed your concerns！

---

### Author Response · Authors · 2022-11-11
**Response to all reviewers:**

We commerce by thanking the five reviewers for their thoughtful and constructive comments. We are really encouraged to see that the reviewers appreciate some positive aspects of our paper, such as an interesting problem to address (Reviewers cfDA, xzm4, PXxs) with strong motivation (Reviewer rozM), good technical novelty (Reviewer si46, cfDA, rozM, yVH3), and promising/effective methodology (Reviewer yVH3, rozM, PXxs). Your expertise significantly helps us strengthen our manuscript – this might be the most helpful review we received in years! In the following parts, we endeavor to provide individual responses to each reviewer. We sincerely hope that a revision (see the updated paper) is still considered.

---

### Author Response · Authors · 2022-11-14
**Response to all reviewers:**

Dear reviewers,

Thank you again for your valuable time and insightful comments. We have provided thorough responses to each reviewer, and sincerely hope you can look through them and update the scores if your concerns have been resolved. We are also open to further discussion if the concerns have not been fully addressed. Please feel free to let us know if you still have any questions.

Best regards!
Authors

---

### Author Response · Authors · 2022-11-24
**Dear Area Chair:**

We thank all reviewers for their thoughtful and constructive reviews, and we have added related content and experiments in the revised manuscript. Meanwhile, we have provided thorough responses to each reviewer and hope you can look through them. We sincerely expect AC to prompt reviewers to actively participate in the discussion to help us make the paper stronger! We are also open to further discussion if the concerns have not been fully addressed.

---

### Decision · Program_Chairs · 2023-01-20

**Decision:**

Accept: poster

**Justification For Why Not Higher Score:**

Both dual LTH and graph LTH were studied before, so this work could be viewed as incremental

**Justification For Why Not Lower Score:**

The authors achieve a triple-win situation of graph lottery tickets with high sparsity, nice performance, and good explainability - which is new. Theory insights, thought not treated particularly as main novel contributions, are offered too

**Metareview: Summary, Strengths And Weaknesses:**

The paper generalizes the Dual Lottery Ticket Hypothesis in GNN network pruning and graph sparsification and proposes a Dual Graph Lottery Ticket framework. Their experimental results demonstrate the effectiveness of their framework. While initially concerns were raised regarding the vague theoretical analysis, unclear training details, and complexity, the authors provide a profound rebuttal that seems to address all reviewers to satisfaction. This paper hence receives a unanimously positive consensus from all reviewers, and AC joins them.

There is a known clarity issue that the authors must immediately fix:  please add a citation to accrediting Wang et al. (2020a) in the part A of the Appendix. The authors should also make more clear that their theory derivations in this part were also inherited from Wang et al. (2020a) - perhaps in order to make the paper more self-contained -  and isn't an independent contribution. This issue was raised and thoroughly investigated during the meta-review process. It is deemed that, since Wang et al. (2020a) was already cited and adequately accredited in the main text on borrowing the GIR regularizer, the credit of prior work was clearly conveyed to reviewers and no confusion was caused. However, Appendix part A should have more strictly followed the citation specification to explicitly indicate the source of the proof idea. The authors are strongly urged to address this concern in their final version.

**Note From Pc:**

if the above contains the word "oral" or "spotlight" please see: "oral" presentation means -> notable-top-5% and "spotlight" means -> notable-top-25%. As stated in our emails, we are disassociating presentation type from AC recommendations